# Better Theory for SGD in the Nonconvex World

**Ahmed Khaled**                                          *ahmed.khaled@princeton.edu*

*Princeton University*

**Peter Richtárik**                                        *peter.richtarik@kaust.edu.sa*
*King Abdullah University of Science and Technology*

**Reviewed on OpenReview:** *https://openreview.net/forum?id=AU4qHN2VkS*

## Abstract

Large-scale nonconvex optimization problems are ubiquitous in modern machine learning, and among practitioners interested in solving them, Stochastic Gradient Descent (SGD) reigns supreme. We revisit the analysis of SGD in the nonconvex setting and propose a new variant of the recently introduced *expected smoothness* assumption which governs the behavior of the second moment of the stochastic gradient. We show that our assumption is both more general and more reasonable than assumptions made in all prior work. Moreover, our results yield the optimal $\mathcal{O}(\varepsilon^{-4})$ rate for finding a stationary point of nonconvex smooth functions, and recover the optimal $\mathcal{O}(\varepsilon^{-1})$ rate for finding a global solution if the Polyak-Łojasiewicz condition is satisfied. We compare against convergence rates under convexity and prove a theorem on the convergence of SGD under Quadratic Functional Growth and convexity, which might be of independent interest. Moreover, we perform our analysis in a framework which allows for a detailed study of the effects of a wide array of sampling strategies and minibatch sizes for finite-sum optimization problems. We corroborate our theoretical results with experiments on real and synthetic data.

## 1 Introduction

In this work we study the *complexity of stochastic gradient descent (SGD)* for solving unconstrained optimization problems of the form

$$\min_{x \in \mathbb{R}^d} f(x), \tag{1}$$

where $f : \mathbb{R}^d \to \mathbb{R}$ is possibly *nonconvex* and satisfies the following smoothness and regularity conditions.

**Assumption 1.** Function $f$ is bounded from below by an infimum $f^{\mathrm{inf}} \in \mathbb{R}$, differentiable, and $\nabla f$ is $L$-Lipschitz: for all $x, y \in \mathbb{R}^d$ we have $\|\nabla f(x) - \nabla f(y)\| \leq L \|x - y\|$.

Motivating this problem is perhaps unnecessary. Indeed, the training of modern deep learning models reduces to nonconvex optimization problems, and the state-of-the-art methods for solving them are all variants of SGD (Salakhutdinov, 2014; Sun, 2020). SGD is a randomized first-order method performing iterations of the form

$$x_{k+1} = x_k - \gamma_k g(x_k), \tag{2}$$

where $g(x)$ is an unbiased estimator of the gradient $\nabla f(x)$ (i.e., $\mathbb{E}[g(x)] = \nabla f(x)$), and $\gamma_k$ is an appropriately chosen learning rate. Since $f$ can have many local minima and/or saddle points, solving (1) to global optimality is intractable (Nemirovsky & Yudin, 1983; Vavasis, 1995). However, the problem becomes tractable if one scales down the requirements on the point of interest from global optimality to some relaxed version thereof, such as stationarity or local optimality. In this paper we are interested in the fundamental problem of finding an $\varepsilon$-stationary point, i.e., we wish to find a random vector $x \in \mathbb{R}^d$ for which $\mathbb{E}\left[\|\nabla f(x)\|^2\right] \leq \varepsilon^2$, where $\mathbb{E}[\cdot]$ is the expectation over the randomness of the algorithm.

### 1.1 Modelling stochasticity

Since unbiasedness alone is not enough to conduct a complexity analysis of SGD, it is necessary to impart further assumptions on the connection between the stochastic gradient $g(x)$ and the true gradient $\nabla f(x)$. The most commonly used assumptions take the form of various structured bounds on the second moment of $g(x)$. We argue (see Section 3) that bounds proposed in the literature are often too strong and unrealistic as they do not fully capture how randomness in $g(x)$ arises in practice. Indeed, existing bounds are primarily constructed in order to facilitate analysis, and their match with reality often takes the back seat. In order to obtain meaningful theoretical insights into the workings of SGD, it is very important to model this randomness both *correctly*, so that the assumptions we impart are provably satisfied, and *accurately*, so as to obtain as tight bounds as possible.

### 1.2 Sources of stochasticity

Practical applications of SGD typically involve the training of supervised machine learning models via empirical risk minimization (Shalev-Shwartz & Ben-David, 2014), which leads to optimization problems of a finite-sum structure:

$$\min_{x \in \mathbb{R}^d} \left( f(x) \stackrel{\text{def}}{=} \frac{1}{n} \sum_{i=1}^{n} f_i(x) \right). \tag{3}$$

In a single-machine setup, $n$ is the number of training data points, and $f_i(x)$ represents the loss of model $x$ on data point $i$. In this setting, data access is expensive, and $g(x)$ is typically constructed via subsampling techniques such as minibatching (Dekel et al., 2012) and importance sampling (Needell et al., 2014). In the rather general arbitrary sampling paradigm (Gower et al., 2019), one may choose an arbitrary random subset $S \subseteq [n]$ of examples, and subsequently $g(x)$ is assembled from the information stored in the gradients $\nabla f_i(x)$ for $i \in S$ only. This leads to formulas of the form

$$g(x) = \sum_{i \in S} v_i \nabla f_i(x), \tag{4}$$

where $v_i$ are appropriately defined random variables ensuring unbiasedness.

In a distributed setting, $n$ corresponds to the number of machines (e.g., number of mobile devices in federated learning) and $f_i(x)$ represents the loss of model $x$ on all the training data stored on machine $i$. In this setting, communication is expensive, and modern gradient-type methods therefore rely on various randomized gradient compression mechanisms such as quantization (Gupta et al., 2015), sparsification (Wangni et al., 2018), and dithering (Alistarh et al., 2017). Given an appropriately chosen (unbiased) randomized compression map $\mathcal{Q} : \mathbb{R}^d \to \mathbb{R}^d$, the local gradients $\nabla f_i(x)$ are first compressed to $\mathcal{Q}_i(\nabla f_i(x))$, where $\mathcal{Q}_i$ is an independent instantiation of $\mathcal{Q}$ sampled by machine $i$ in each iteration, and subsequently communicated to a master node, which performs aggregation (Khirirat et al., 2018). This gives rise to SGD with stochastic gradient of the form

$$g(x) = \frac{1}{n} \sum_{i=1}^{n} \mathcal{Q}_i(\nabla f_i(x)). \tag{5}$$

In many applications, each $f_i$ has a finite sum structure of its own, reflecting the empirical risk composed of the training data stored on that device. In such situations, it is often assumed that compression is not applied to exact gradients, but to stochastic gradients coming from subsampling (Gupta et al., 2015; Ben-Nun & Hoefler, 2019; Horváth et al., 2019). This further complicates the structure of the stochastic gradient.

## 2 Contributions

The highly specific and elaborate structure of the stochastic gradient $g(x)$ used in practice, such as that coming from subsampling as in (4) or compression as in (5), raises questions about appropriate theoretical modelling of its second moment. As we shall explain in Section 3, existing approaches do not offer a satisfactory treatment.

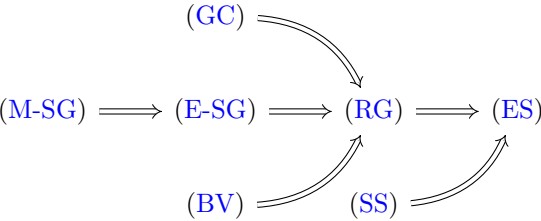

Figure 1: Assumption hierarchy on the second moment of the stochastic gradient of nonconvex functions. An arrow indicates implication. Our newly proposed ES assumption is the most general. The statement is formalized as Theorem 1.

Indeed, as we show through a simple example, none of the existing assumptions are satisfied even in the simple scenario of subsampling from the sum of two functions (see Proposition 1).

Our work is motivated by the need of a more accurate modelling of the stochastic gradient for nonconvex optimization problems, which we argue would lead to a more accurate and informative analysis of SGD in the nonconvex world–a problem of the highest importance in modern deep learning.

The key contributions of our work are:

- Inspired by recent developments in the analysis of SGD in the (strongly) convex setting (Richtárik & Takáč, 2017; Gower et al., 2020; 2019), we propose a new[1] assumption, which we call *expected smoothness (ES)*, for modelling the second moment of the stochastic gradient, specifically focusing on nonconvex problems (see Section 4). In particular, we assume that there exist constants $A, B, C \geq 0$ such that
$$\mathbb{E}\left[\|g(x)\|^2\right] \leq 2A\left(f(x) - f^{\text{inf}}\right) + B\|\nabla f(x)\|^2 + C.$$

- We show in Section 4.3 that (ES) is the weakest, and hence the most general, among all assumptions in the existing literature we are aware of (see Figure 1), including assumptions such as bounded variance (BV) (Ghadimi & Lan, 2013), maximal strong growth (M-SG) (Schmidt & Roux, 2013), strong growth (SG) (Vaswani et al., 2019), relaxed growth (RG) (Bottou et al., 2018), and gradient confusion (GC) (Sankararaman et al., 2020), which we review in Section 3.

- Moreover, we prove that unlike existing assumptions, which typically implicitly assume that stochasticity comes from perturbation (see Section 4.4), (ES) automatically holds under standard and weak assumptions made on the loss function in settings such as *subsampling* (see Section 4.5) and *compression* (see Section 4.6). In this sense, (ES) not an assumption but an inequality which provably holds and can be used to accurately and more precisely capture the convergence of SGD. For instance, to the best of our knowledge, while the combination of gradient compression and subsampling is not covered by any prior analysis of SGD for nonconvex objectives, our results can be applied to this setting.

- We recover the optimal $\mathcal{O}(\varepsilon^{-4})$ rate for general smooth nonconvex problems and $\mathcal{O}(\varepsilon^{-1})$ rate under the PL condition (see Section 5). However, our rates are informative enough for us to be able to deduce, for the first time in the literature on nonconvex SGD, importance sampling probabilities and formulas for the optimal minibatch size (see Section 6).

## 3 Existing Models of Stochastic Gradient

Ghadimi & Lan (2013) analyze SGD under the assumption that $f$ is lower bounded, that the stochastic gradients $g(x)$ are unbiased and have **bounded variance**

$$\mathbb{E}\left[\|g(x) - \nabla f(x)\|^2\right] \leq \sigma^2. \tag{BV}$$

---

[1]After writing this manuscript, we became aware of the early work (Polyak & Tsypkin, 1973) that had a similar assumption. We discuss this in Section 4.

Note that due to unbiasedness, this is equivalent to

$$\mathbb{E}\left[\|g(x)\|^2\right] \leq \|\nabla f(x)\|^2 + \sigma^2. \tag{6}$$

With an appropriately chosen constant stepsize $\gamma$, their results imply a $\mathcal{O}(\varepsilon^{-4})$ rate of convergence.

In the context of finite-sum problems with uniform sampling, where at each step one $i \in [n]$ is sampled uniformly and the stochastic gradient estimator used is $g(x) = \nabla f_i(x)$ for a randomly selected index $i$, the **maximal strong growth** condition requires the inequality

$$\|g(x)\|^2 \leq \alpha \|\nabla f(x)\|^2 \tag{M-SG}$$

to hold almost surely for some $\alpha \geq 0$. Tseng (1998) used the maximal strong growth condition early to establish the convergence of the incremental gradient method, a closely related variant of SGD. Schmidt & Roux (2013) prove the linear convergence of SGD for strongly convex objectives under (M-SG).

We may also assume that (M-SG) holds in expectation rather than uniformly, leading to **expected strong growth**:

$$\mathbb{E}\left[\|g(x)\|^2\right] \leq \alpha \|\nabla f(x)\|^2 \tag{E-SG}$$

for some $\alpha > 0$. Like maximal growth, variants of this condition have also been used in convergence results for incremental gradient methods (Solodov, 1998). Vaswani et al. (2019) prove that under (E-SG), SGD converges to an $\varepsilon$-stationary point in $\mathcal{O}(\varepsilon^{-2})$ steps. This assumption is quite strong and necessitates interpolation: if $\nabla f(x) = 0$, then $g(x) = 0$ almost surely. This is typically not true in the distributed, finite-sum case where the functions $f_i$ can be very different; see e.g., (McMahan et al., 2017).

Bottou et al. (2018) consider the **relaxed growth condition** which is a version of (E-SG) featuring an additive constant:

$$\mathbb{E}\left[\|g(x)\|^2\right] \leq \alpha \|\nabla f(x)\|^2 + \beta. \tag{RG}$$

In view of (6), (RG) can be also seen as a slight generalization of the bounded variance assumption (BV). Bertsekas & Tsitsiklis (2000) establish the almost-sure convergence of SGD under (RG), and Bottou et al. (2018) give a $\mathcal{O}(\varepsilon^{-2})$ convergence rate for nonconvex objectives to a neighborhood of a stationary point of radius linearly proportional to $\beta$. Unfortunately, (RG) is quite difficult to verify in practice and can be shown not to hold for some simple problems, as the following proposition shows.

**Proposition 1.** There is a simple finite-sum minimization problem with two functions for which (RG) is not satisfied.

The proof of this proposition and all subsequent proofs are relegated to the supplementary material.

In recent development, Sankararaman et al. (2020) postulate a **gradient confusion** bound for finite-sum problems and SGD with uniform single-element sampling. This bound requires the existence of $\eta > 0$ such that

$$\langle \nabla f_i(x), \nabla f_j(x) \rangle \geq -\eta \tag{GC}$$

holds for all $i \neq j$ (and all $x \in \mathbb{R}^d$). For general nonconvex objectives, they prove convergence to a neighborhood only. For functions satisfying the PL condition (Assumption 5), they prove linear convergence to a neighborhood of a stationary point.

Lei et al. (2019) analyze SGD for $f$ of the form

$$f(x) = \mathbb{E}_{\xi \sim \mathcal{D}}\left[f_\xi(x)\right]. \tag{7}$$

They assume that $f_\xi$ is nonnegative and almost-surely $\alpha$-Hölder-continuous in $x$ and then use $g(x) = \nabla f_\xi(x)$ for $\xi$ sampled i.i.d. from $\mathcal{D}$. Specialized to the case of $L$-smoothness, their assumption reads

$$\|g(x) - g(y)\| \leq L \|x - y\| \quad \text{and} \quad f_\xi(x) \geq 0 \tag{SS}$$

almost surely for all $x, y \in \mathbb{R}^d$. We term this condition **sure-smoothness** (SS). They establish the sample complexity $\mathcal{O}(\varepsilon^{-4} \log(\varepsilon^{-1}))$. Unfortunately, their results do not recover full gradient descent and their analysis is not easily extendable to compression and subsampling.

We summarize the relations between the aforementioned assumptions and our own (introduced in Section 4) in Figure 1. These relations are formalized as Theorem 1.

## 4 ES in the Nonconvex World

In this section, we first briefly review the notion of expected smoothness as recently proposed in several contexts different from ours, and use this development to motivate our definition of expected smoothness (ES) for nonconvex problems. We then proceed to show that (ES) is the weakest from all previous assumptions modelling the behavior of the second moment of stochastic gradient for nonconvex problems reviewed in Section 3, thus substantiating Figure 1. Finally, we show how (ES) provides a *correct* and *accurate* model for the behavior of the stochastic gradient arising not only from classical perturbation, but also from subsampling and compression.

### 4.1 Brief history of expected smoothness: from convex quadratic to convex optimization

Our starting point is the work of Richtárik & Takáč (2017) who, motivated by the desire to obtain deeper insights into the workings of the sketch and project algorithms developed by Gower & Richtárik (2015), study the behavior of SGD applied to a reformulation of a consistent linear system as a stochastic convex quadratic optimization problem of the form

$$\min_{x \in \mathbb{R}^d} f(x) \stackrel{\text{def}}{=} \mathbb{E}\left[f_\xi(x)\right]. \tag{8}$$

The above problem encodes a linear system in the sense that $f(x)$ is nonnegative and equal to zero if and only if $x$ solves the linear system. The distribution behind the randomness in their reformulation (8) plays the role of a parameter which can be tuned in order to target specific properties of SGD, such as convergence rate or cost per iteration. The stochastic gradient $g(x) = \nabla f_\xi(x)$ satisfies the identity $\mathbb{E}\left[\|g(x)\|^2\right] = 2f(x)$, which plays a key role in the analysis. Since in their setting $g(x_\star) = 0$ almost surely for any minimizer $x_\star$ (which suggests that their problem is over-parameterized), the above identity can be written in the equivalent form

$$\mathbb{E}\left[\|g(x) - g(x_\star)\|^2\right] = 2\left(f(x) - f(x_\star)\right). \tag{9}$$

Equation (9) is the first instance of the *expected smoothness* property/inequality we are aware of. Using tools from matrix analysis, Richtárik & Takáč (2017) are able to obtain identities for the expected iterates of SGD. Kovalev et al. (2018) study the same method for *spectral* distributions and for some of these establish stronger *identities* of the form $\mathbb{E}\left[\|x_k - x_\star\|^2\right] = \alpha^k \|x_0 - x_\star\|^2$, suggesting that the property (9) has the capacity to achieve a perfectly precise mean-square analysis of SGD in this setting.

Expected smoothness as an inequality was later used to analyze the JacSketch method (Gower et al., 2020), which is a general variance-reduced SGD that includes the widely-used SAGA algorithm (Defazio et al., 2014) as a special case. By carefully considering and optimizing over smoothness, Gower et al. (2020) obtain the currently best-known convergence rate for SAGA. Assuming strong convexity and the existence a global minimizer $x_\star$, their assumption in our language reads

$$\mathbb{E}\left[\|g(x) - g(x_\star)\|^2\right] \leq 2A\left(f(x) - f(x_\star)\right), \tag{C-ES}$$

where $A \geq 0$, $g(x)$ is a stochastic gradient and the expectation is w.r.t. the randomness embedded in $g$. We refer to the above condition by the name *convex expected smoothness* (C-ES) as it provides a good model of the stochastic gradient of convex objectives. (C-ES) was subsequently used to analyze SGD for quasi strongly-convex functions by Gower et al. (2019), which allowed the authors to study a wide array of subsampling strategies with great accuracy as well as provide the first formulas for the optimal minibatch size

for SGD in the strongly convex regime. These rates are tight up to non-problem specific constants in the setting of convex stochastic optimization (Nguyen et al., 2019).

Independently and motivated by extending the analysis of subgradient methods for convex objectives, Grimmer (2019) also studied the convergence of SGD under a similar setting to (C-ES) (in fact, their assumption is (10) below). Grimmer (2019) develops quite general and elegant theory that includes the use of projection operators and non-smooth functions, but the convergence rate they obtain for smooth strongly convex functions shows a suboptimal dependence on the condition number.

## 4.2 Expected smoothness for nonconvex optimization

Given the utility of (C-ES), it is then natural to ask: *can we extend* (C-ES) *beyond convexity?* The first problem we are faced with is that $x_\star$ is ill-defined for nonconvex optimization problems, which may not have any global minima. However, Gower et al. (2019) use (C-ES) through following direct consequence of (C-ES) only:

$$\mathbb{E}\left[\|g(x)\|^2\right] \leq 4A\left(f(x) - f(x_\star)\right) + 2\sigma^2, \tag{10}$$

where $\sigma^2 \stackrel{\text{def}}{=} \mathbb{E}\left[\|g(x_\star)\|^2\right]$. We thus propose to remove $x_\star$ and merely ask for a global lower bound $f^{\text{inf}}$ on the function $f$ rather than a global minimizer, dispense with the interpretation of $\sigma^2$ as the variance at the optimum $x_\star$ and merely ask for the existence of any such constant. This yields the new condition

$$\mathbb{E}\left[\|g(x)\|^2\right] \leq 4A\left(f(x) - f^{\text{inf}}\right) + C, \tag{11}$$

for some $A, C \geq 0$. While (11) may be satisfactory for the analysis of convex problems, it does not enable us to easily recover the convergence of full gradient descent or SGD under strong growth in the case of nonconvex objectives. As we shall see in further sections, the fix is to add a third term to the bound, which finally leads to our (ES) assumption:

**Assumption 2** (Expected smoothness)**.** The second moment of the stochastic gradient satisfies

$$\mathbb{E}\left[\|g(x)\|^2\right] \leq 2A\left(f(x) - f^{\text{inf}}\right) + B \cdot \|\nabla f(x)\|^2 + C, \tag{ES}$$

for some $A, B, C \geq 0$ and all $x \in \mathbb{R}^d$.

After we wrote this manuscript, we became aware of the very early work (Polyak & Tsypkin, 1973). They considered a similar assumption while analyzing pseudogradient algorithms, and they established an asymptotic convergence bound for a variant of gradient descent. We arrived at this assumption independently and via the different motivation outlined above, and our proof techniques are aimed towards finding non-asymptotic convergence rates. We believe the rediscovery of this assumption is testimony to its utility, and our work significantly extends this early work with non-asymptotic convergence rates in several settings and other new contributions. Appendix H gives a more detailed overview of the assumption used by Polyak & Tsypkin (1973).

In the rest of this section, we turn to the generality of Assumption 2 and its use in *correct* and *accurate* modelling of sources of stochasticity arising in practice.

## 4.3 Expected smoothness as the weakest assumption

As discussed in Section 3, assumptions on the stochastic gradients abound in the literature on SGD. If we hope for a correct and tighter theory, then we may ask that we at least recover the convergence of SGD under those assumptions. Our next theorem, described informally below and stated and proved formally in the supplementary material, proves exactly this.

**Theorem 1** (Informal)**.** The expected smoothness assumption (Assumption 2) is the weakest condition among the assumptions in Section 3.

### 4.4 Perturbation

One of the simplest models of stochasticity is the case of additive zero-mean noise with bounded variance, that is

$$g(x) = \nabla f(x) + \xi,$$

where $\xi$ is a random variable satisfying $\mathbb{E}[\xi] = 0$ and $\mathbb{E}\left[\|\xi\|^2\right] \leq \sigma^2$. Because the bounded variance condition (BV) is clearly satisfied, then by Theorem 1, our Assumption 2 is also satisfied. While this model can be useful for modelling artificially injected noise into the full gradient (Ge et al., 2015; Fang et al., 2019), it is unreasonably strong for practical sources of noise: indeed, as we saw in Proposition 1, it does not hold for subsampling with just two functions. It is furthermore unable to model sources of rather simple *multiplicative noise* which arises in the case of gradient compression operators.

### 4.5 Subsampling

Now consider $f$ having a finite-sum structure (3). In order to develop a general theory of SGD for a wide array of subsampling strategies, we follow the *stochastic reformulation* formalism pioneered by Richtárik & Takáč (2017); Gower et al. (2020) in the form proposed in (Gower et al., 2019). Given a sampling vector $v \in \mathbb{R}^n$ drawn from some user-defined distribution $\mathcal{D}$ (where a sampling vector is one such that $\mathbb{E}_{\mathcal{D}}[v_i] = 1$ for all $i \in [n]$), we define the random function $f_v(x) \stackrel{\text{def}}{=} \frac{1}{n} \sum_{i=1}^{n} v_i f_i(x)$. Noting that $\mathbb{E}_{v \sim \mathcal{D}}[f_v(x)] = f(x)$, we reformulate (3) as a stochastic minimization problem

$$\min_{x \in \mathbb{R}^d} \left( \mathbb{E}_{v \sim \mathcal{D}}[f_v(x)] = \mathbb{E}_{v \sim \mathcal{D}}\left[\frac{1}{n} \sum_{i=1}^{n} v_i f_i(x)\right]\right), \tag{12}$$

where we assume only access to unbiased estimates of $\nabla f(x)$ through the stochastic realizations

$$\nabla f_v(x) = \frac{1}{n} \sum_{i=1}^{n} v_i \nabla f_i(x). \tag{13}$$

That is, given current point $x$, we sample $v \sim \mathcal{D}$ and set $g(x) \stackrel{\text{def}}{=} \nabla f_v(x)$. We will now show that (ES) is satisfied under very mild and natural assumptions on the functions $f_i$ and the sampling vectors $v_i$. In that sense, (ES) is not an additional *assumption*, it is an *inequality* that is automatically satisfied.

**Assumption 3.** Each $f_i$ is bounded from below by $f_i^{\text{inf}}$ and is $L_i$-smooth: That is, for all $x, y \in \mathbb{R}^d$ we have

$$f_i(y) \leq f_i(x) + \langle \nabla f_i(x), y - x \rangle + \frac{L_i}{2} \|y - x\|^2.$$

To show that Assumption 2 is an automatic consequence of Assumption 3, we rely on the following crucial lemma.

**Lemma 1.** Let $f$ be a function for which Assumption 1 is satisfied. Then for all $x \in \mathbb{R}^d$ we have

$$\|\nabla f(x)\|^2 \leq 2L\left(f(x) - f^{\text{inf}}\right). \tag{14}$$

This lemma shows up in several recent works and is often used in conjunction with other assumptions such as bounded variance (Li & Orabona, 2019) and convexity (Stich & Karimireddy, 2019). Lei et al. (2019) also use a version of it to prove the convergence of SGD for nonconvex objectives, and we compare our results against theirs in Section 5. Armed with Lemma 1, we can prove that Assumption 2 holds for all non-degenerate distributions $\mathcal{D}$.

**Proposition 2.** Suppose that Assumption 3 holds and that $\mathbb{E}\left[v_i^2\right]$ is finite for all $i$. Let $\Delta^{\text{inf}} \stackrel{\text{def}}{=} \frac{1}{n} \sum_{i=1}^{n} f^{\text{inf}} - f_i^{\text{inf}}$. Then $\Delta^{\text{inf}} \geq 0$ and Assumption 2 holds with $A = \max_i L_i \mathbb{E}\left[v_i^2\right]$, $B = 0$ and $C = 2A\Delta^{\text{inf}}$.

The condition that $\mathbb{E}\left[v_i^2\right]$ is finite is a very mild condition on $\mathcal{D}$ and is satisfied for virtually all practical subsampling schemes in the literature. However, the generality of Proposition 2 comes at a cost: the bounds are too pessimistic. By making more specific (and practical) choices of the sampling distribution $\mathcal{D}$, we can get much tighter bounds. We do this by considering some representative sampling distributions next, without aiming to be exhaustive:

- **Sampling with replacement.** An $n$-sided die is rolled a total of $\tau > 0$ times and the number of times the number $i$ shows up is recorded as $S_i$. We can then define

$$v_i = \frac{S_i}{\tau q_i}, \tag{15}$$

  where $q_i$ is the probability that the $i$-th side of the die comes up and $\sum_{i=1}^n q_i = 1$. In this case, the number of stochastic gradients queried is always $\tau$ though some of them may be repeated.

- **Independent sampling without replacement.** We generate a random subset $S \subseteq \{1, 2, \ldots, n\}$ and define

$$v_i = \frac{1_{i \in S}}{p_i}, \tag{16}$$

  where $1_{i \in S} = 1$ if $i \in S$ and 0 otherwise, and $p_i = \text{Prob}\,(i \in S) > 0$. We assume that each number $i$ is included in $S$ with probability $p_i$ independently of all the others. In this case, the number of stochastic gradients queried $|S|$ is not fixed but has expectation $\mathbb{E}\left[|S|\right] = \sum_{i=1}^n p_i$.

- **$\tau$-nice sampling without replacement.** This is similar to the previous sampling, but we generate a random subset $S \subseteq \{1, 2, \ldots, n\}$ by choosing uniformly from all subsets of size $\tau$ for integer $\tau \in [1, n]$. We define $v_i$ as in (16) and it is easy to see that $p_i = \frac{\tau}{n}$ for all $i$.

These sampling distributions were considered in the context of SGD for convex objective functions in (Gorbunov et al., 2020; Gower et al., 2019). We show next that Assumption 2 is satisfied for these distributions with much better constants than the generic Proposition 2 would suggest.

**Proposition 3.** Suppose that Assumptions 1 and 3 hold and let $\Delta^{\text{inf}} = \frac{1}{n} \sum_{i=1}^n (f^{\text{inf}} - f_i^{\text{inf}})$. Then:

(i) For independent sampling with replacement, Assumption 2 is satisfied with $A = \max_i \frac{L_i}{\tau n q_i}$, $B = 1 - \frac{1}{\tau}$, and $C = 2A\Delta^{\text{inf}}$.

(ii) For independent sampling without replacement, Assumption 2 is satisfied with $A = \max_i \frac{(1-p_i)L_i}{p_i n}$, $B = 1$, and $C = 2A\Delta^{\text{inf}}$.

(iii) For $\tau$-nice sampling without replacement, Assumption 2 is satisfied with $A = \frac{n-\tau}{\tau(n-1)} \max_i L_i$, $B = \frac{n(\tau-1)}{\tau(n-1)}$, and $C = 2A\Delta^{\text{inf}}$.

### 4.6 Compression

We now further show that our framework is general enough to capture the convergence of stochastic gradient quantization or compression schemes. Consider the finite-sum problem (3) and let $g_i(x)$ be stochastic gradients such that $\mathbb{E}\left[g_i(x)\right] = \nabla f_i(x)$. We construct an estimator $g(x)$ via

$$g(x) = \frac{1}{n} \sum_{i=1}^n \mathcal{Q}_i(g_i(x)), \tag{17}$$

where the $\mathcal{Q}_i$ are sampled independently for all $i$ and across all iterations. Clearly, this generalizes (5). We consider the class of $\omega$-*compression operators*:

**Assumption 4.** We say that a stochastic operator $\mathcal{Q} = \mathcal{Q}_\xi : \mathbb{R}^d \to \mathbb{R}^d$ is an $\omega$-compression operator if

$$\mathbb{E}_\xi\left[\mathcal{Q}_\xi(x)\right] = x, \quad \mathbb{E}_\xi\left[\|\mathcal{Q}_\xi(x) - x\|^2\right] \le \omega\|x\|^2. \tag{18}$$

Assumption 4 is mild and is satisfied by many compression operators in the literature, including random dithering (Alistarh et al., 2017), random sparsification, block quantization (Horváth et al., 2019), and others. The next proposition then shows that if the stochastic gradients $g_i(x)$ themselves satisfy Assumption 2 with their respective functions $f_i$, then $g(x)$ also satisfies Assumption 2.

**Proposition 4.** Suppose that a stochastic gradient estimator $g(x)$ is constructed via (17) such that each $\mathcal{Q}_i$ is a $\omega_i$-compressor satisfying Assumption 4. Suppose further that the stochastic gradient $g_i(x)$ is such that $\mathbb{E}\left[g_i(x)\right] = \nabla f_i(x)$ and that each satisfies Assumption 2 with constants $(A_i, B_i, C_i)$. Then there exists constants $A, B, C \geq 0$ such that $g(x)$ satisfies Assumption 2 with $(A, B, C)$.

To the best of our knowledge, the combination of gradient compression and subsampling is not covered by any analysis of SGD for nonconvex objectives. Hence, Proposition 4 shows that Assumption 2 is indeed versatile enough to model practical and diverse sources of stochasticity well.

## 5 SGD in the Nonconvex World

### 5.1 General convergence theory

Our main convergence result relies on the following key lemma.

**Lemma 2.** Suppose that Assumptions 1 and 2 are satisfied. Choose constant stepsize $\gamma > 0$ such that $\gamma \leq \frac{1}{BL}$. Then,

$$\frac{1}{2} \sum_{k=0}^{K-1} w_k r_k + \frac{w_{K-1}}{\gamma} \delta_K \leq \frac{w_{-1}}{\gamma} \delta_0 + \frac{LC}{2} \sum_{k=0}^{K-1} w_k \gamma.$$

where $r_k \stackrel{\text{def}}{=} \mathbb{E}\left[\|\nabla f(x_k)\|^2\right]$, $w_k \stackrel{\text{def}}{=} \frac{w_{-1}}{(1+L\gamma^2 A)^{k+1}}$ for $w_{-1} > 0$ arbitrary, and $\delta_k \stackrel{\text{def}}{=} \mathbb{E}\left[f(x_k)\right] - f^{\text{inf}}$.

Lemma 2 bounds a *weighted sum* of stochastic gradients over the entire run of the algorithm. This idea of weighting different iterates has been used in the analysis of SGD in the convex case (Rakhlin et al., 2012; Shamir & Zhang, 2013; Stich, 2019) typically with the goal of returning a weighted average of the iterates $\bar{x}_K$ at the end. In contrast, we only use the weighting to facilitate the proof.

**Theorem 2.** Suppose that Assumptions 1 and 2 hold. Suppose that a stepsize $\gamma > 0$ is chosen such that $\gamma \leq \frac{1}{LB}$. Letting $\delta_0 \stackrel{\text{def}}{=} f(x_0) - f^{\text{inf}}$, we have

$$\min_{0 \leq k \leq K-1} \mathbb{E}\left[\|\nabla f(x_k)\|^2\right] \leq LC\gamma + \frac{2\left(1 + L\gamma^2 A\right)^K}{\gamma K} \delta_0.$$

While the bound of Theorem 2 shows possible exponential *blow-up*, we can show that by carefully controlling the stepsize we can nevertheless attain an $\varepsilon$-stationary point given $\mathcal{O}(\varepsilon^{-4})$ stochastic gradient evaluations. This dependence is in fact *optimal* for SGD without extra assumptions on second-order smoothness or disruptiveness of the stochastic gradient noise (Drori & Shamir, 2020). We use a similar stepsize to Ghadimi & Lan (2013).

**Corollary 1.** Fix $\varepsilon > 0$. Choose the stepsize $\gamma > 0$ as $\gamma = \min\left\{\frac{1}{\sqrt{LAK}}, \frac{1}{LB}, \frac{\varepsilon}{2LC}\right\}$. Then provided that

$$K \geq \frac{12\delta_0 L}{\varepsilon^2} \max\left\{B, \frac{12\delta_0 A}{\varepsilon^2}, \frac{2C}{\varepsilon^2}\right\}, \tag{19}$$

we have $\min_{0 \leq k \leq K-1} \mathbb{E}\left[\|\nabla f(x_k)\|\right] \leq \varepsilon$.

As a start, the iteration complexity given by (19) recovers full gradient descent: plugging in $B = 1$ and $A = C = 0$ shows that we require a total of $12\delta_0 L\varepsilon^{-2}$ iterations in required to a reach an $\varepsilon$-stationary point. This is the standard rate of convergence for gradient descent on nonconvex objectives (Beck, 2017), up to absolute (non-problem-specific) constants.

Plugging in $A = C = 0$ and $B$ to be any nonnegative constant recovers the fast convergence of SGD under strong growth (E-SG). Our bounds are similar to Lei et al. (2019) but improve upon them by recovering full gradient descent, assuming smoothness only in expectation, and attaining the optimal $\mathcal{O}(\varepsilon^{-4})$ rate without logarithmic terms.

## 5.2 Convergence under the Polyak-Łojasiewicz condition

One of the popular generalizations of strong convexity in the literature is the Polyak-Łojasiewicz (PL) condition (Karimi et al., 2016; Lei et al., 2019). We first define this condition and then establish convergence of SGD for functions satisfying it and our (ES) assumption. In the rest of this section, we assume that the function $f$ has a minimizer and denote $f^\star \overset{\text{def}}{=} \min f$.

**Assumption 5.** We say that a differentiable function $f$ satisfies the Polyak-Łojasiewicz condition if for all $x \in \mathbb{R}^d$,

$$\frac{1}{2}\|\nabla f(x)\|^2 \geq \mu\left(f(x) - f^\star\right).$$

We rely on the following lemma where we use the stepsize sequence recently introduced by Stich (2019) but without iterate averaging, as averaging in general may not make sense for nonconvex models.

**Lemma 3.** Consider a sequence $(r_t)_t$ satisfying

$$r_{t+1} \leq (1 - a\gamma_t) r_t + c\gamma_t^2, \tag{20}$$

where $\gamma_t \leq \frac{1}{b}$ for all $t \geq 0$ and $a, c \geq 0$ with $a \leq b$. Fix $K > 0$ and let $k_0 = \lceil \frac{K}{2} \rceil$. Then choosing the stepsize as

$$\gamma_t = \begin{cases} \frac{1}{b}, & \text{if } K \leq \frac{b}{a} \text{ or } t < k_0, \\ \frac{2}{a(s+t-k_0)} & \text{if } K \geq \frac{b}{a} \text{ and } t > k_0 \end{cases}$$

with $s = \frac{2b}{a}$ gives $r_K \leq \exp\left(-\frac{aK}{2b}\right) r_0 + \frac{9c}{a^2 K}$.

Using the stepsize scheme of Lemma 3, we can show that SGD finds an optimal global solution at a $1/K$ rate, where $K$ is the total number of iterations.

**Theorem 3.** Suppose that Assumptions 1, 2, and 5 hold. Suppose that SGD is run for $K > 0$ iterations with the stepsize sequence $(\gamma_k)_k$ of Lemma 3 with $\gamma_k \leq \min\left\{\frac{\mu}{2AL}, \frac{1}{2BL}\right\}$ for all $k$. Then

$$\mathbb{E}\left[f(x_K) - f^*\right] \leq \frac{9\kappa_f C}{2\mu K} + \exp\left(-\frac{K}{2\kappa_f \max\{\kappa_S, B\}}\right)\left(f(x_0) - f^*\right).$$

where $\kappa_f \overset{\text{def}}{=} L/\mu$ is the condition number of $f$ and $\kappa_S \overset{\text{def}}{=} A/\mu$ is the stochastic condition number.

The next corollary recovers the $\mathcal{O}(\varepsilon^{-1})$ convergence rate for strongly convex functions, which is the optimal dependence on the accuracy $\varepsilon$ (Nguyen et al., 2019).

**Corollary 2.** In the same setting of Theorem 3, fix $\varepsilon > 0$. Let $r_0 \overset{\text{def}}{=} f(x_0) - f^\star$. Then $\mathbb{E}\left[f(x_K) - f^\star\right] \leq \varepsilon$ as long as

$$K \geq \kappa_f \max\left\{2\kappa_S \log\left(\frac{2r_0}{\varepsilon}\right), 2B \log\left(\frac{2r_0}{\varepsilon}\right), \frac{9C}{2\mu\varepsilon}\right\}.$$

While the dependence on $\varepsilon$ is optimal, the situation is different when we consider the dependence on problem constants, and in particular the dependence on $\kappa_S$: Corollary 2 shows a possibly *multiplicative* dependence $\kappa_f \kappa_S$. This is different for objectives where we assume convexity, and we show this next. It is known that the PL-condition implies the *quadratic functional growth* (QFG) condition (Necoara et al., 2019), and is in fact equivalent to it for convex and smooth objectives (Karimi et al., 2016). We will adopt this assumption in conjunction with the convexity of $f$ for our next result.

**Assumption 6.** We say that a convex function $f$ satisfies the quadratic functional growth condition if

$$f(x) - f^\star \geq \frac{\mu}{2}\|x - \pi(x)\|^2 \tag{21}$$

for all $x \in \mathbb{R}^d$ where $f^\star$ is the minimum value of $f$ and where $\pi(x) \stackrel{\text{def}}{=} \arg\min_{y \in X^\star} \|x - y\|^2$ is the projection on the set of minima $X^\star \stackrel{\text{def}}{=} \{y \in \mathbb{R}^d \mid f(y) = f^\star\}$.

There are only a handful of results under QFG (Drusvyatskiy & Lewis, 2018; Necoara et al., 2019; Grimmer, 2019) and only one applies to our setting (Grimmer, 2019). For QFG in conjunction with convexity and expected smoothness, we can prove convergence in function values similar to Theorem 3,

**Theorem 4.** Assume that Assumptions 2 and 6 hold with $f$ convex. Choose $\gamma \leq \min\left\{\frac{1}{4L}, \frac{1}{4(BL+A)}\right\}$ according to Lemma 3. Then

$$\mathbb{E}\left[f(x_K) - f^\star\right] \leq \frac{18\kappa_f C}{\mu K} + \frac{\kappa_f}{2}\exp\left(-\frac{K}{M}\right)(f(x_0) - f^\star),$$

where $M \stackrel{\text{def}}{=} 8\max\{4\kappa_f, 4B\kappa_f + \kappa_S\}$.

Theorem 4 allows stepsizes $\mathcal{O}(1/L)$, which are much larger than the $\mathcal{O}(1/(\kappa L))$ stepsizes in (Nguyen et al., 2019; Grimmer, 2019). Hence, it improves upon the prior results of Nguyen et al. (2019) and Grimmer (2019) in the context of finite-sum problems where the individual functions $f_i$ are smooth but possibly nonconvex, and the average $f$ is strongly convex or satisfies Assumption 6.

The following straightforward corollary of Theorem 4 shows that when convexity is assumed, we can get a dependence on the *sum* of the condition numbers $\kappa_f + \kappa_S$ rather than their product. This is a significant difference from the nonconvex setting, and it is not known whether it is an artifact of our analysis or an inherent difference.

**Corollary 3.** In the same setting of Theorem 4, fix $\varepsilon > 0$. Then $\mathbb{E}\left[f(x_K) - f^\star\right] \leq \epsilon$ as long as

$$K \geq \max\left\{\frac{36\kappa_f C}{\mu\epsilon}, M\log\left(\frac{4\kappa_f r_0}{\epsilon}\right)\right\},$$

where $M \stackrel{\text{def}}{=} 32B\kappa_f + 8\kappa_S$ and $r_0 = f(x_0) - f^\star$.

## 6 Importance Sampling and Optimal Minibatch Size

As an example application of our results, we consider *importance sampling*: choosing the sampling distribution to maximize convergence speed. We consider independent sampling with replacement with minibatch size $\tau$. Plugging the bound on $A, B, C$ from Proposition 3 into the sample complexity from Corollary 1 yields:

$$K \geq \frac{12\delta_0 L}{\varepsilon^2}\max\left\{\left(1 - \tau^{-1}\right), \frac{D}{\varepsilon^2}\max_i \frac{L_i}{\tau n q_i}\right\}, \tag{22}$$

where $D = \max\left\{12\delta_0, 4\Delta^{\inf}\right\}$. Optimizing (22) over $(q_i)_{i=1}^n$ yields the sampling distribution

$$q_i^\star = \frac{L_i}{\sum_{j=1}^n L_j}. \tag{23}$$

The same sampling distribution has appeared in the literature before (Zhao & Zhang, 2015; Needell et al., 2014), and our work is the first to give it justification for SGD on nonconvex objectives. Plugging the distribution of (23) into (22) and considering the total number of stochastic gradient evaluations $K \times \tau$ we get,

$$K\tau \geq \frac{12\delta_0 L}{\varepsilon^2}\max\left\{\tau - 1, \frac{D\bar{L}}{\varepsilon^2}\right\}$$

where $\bar{L} \stackrel{\text{def}}{=} \frac{1}{n}\sum_{i=1}^n L_i$. This is minimized over the minibatch size $\tau$ whenever $\tau \leq \tau^* = 1 + \lfloor D\bar{L}\varepsilon^{-2}\rfloor$. Similar expressions for importance sampling and minibatch size can be obtained for other sampling distributions as in (Gower et al., 2019).

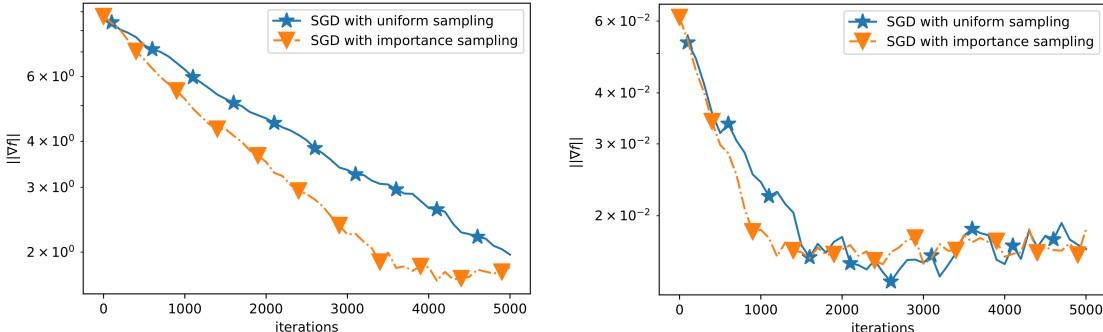

Figure 2: Results on regularized linear regression with (right) and without (left) normalization. Normalization means forcing $\|a_i\| = 1$.

Table 1: Fitted constants in the regularized logistic regression problem. Predicted constants are per Proposition 3. The residual is the mean square error, see Section G.1 in the supplementary for more discussion of this table.

|  | $2A$ | $B$ | $C$ | Residual |
|---|---|---|---|---|
| ES, Predicted | 9 | 0 | 0.994 | 6.113 |
| ES, Fit | 10.09 | 0 | 0.373 | 0.413 |
|  | - | $\alpha$ | $\beta$ | Residual |
| RG, Fit | - | 0.38 | 2.09 | 0.57 |

## 7 Experiments

### 7.1 Linear regression with a nonconvex regularizer

We first consider a linear regression problem with nonconvex regularization to test the importance sampling scheme given in Section 6,

$$\min_{x \in \mathbb{R}^d} \left\{ f(x) \stackrel{\text{def}}{=} \frac{1}{n} \sum_{i=1}^{n} \left( l_i(x) + \lambda \sum_{j=1}^{d} \frac{x_i^2}{1 + x_i^2} \right) \right\}, \tag{24}$$

where $l_i(x) = \|\langle a_i, x \rangle - y_i\|^2$, $a_1, \ldots, a_n \in \mathbb{R}^d$ are generated, and $\lambda = 0.1$. We use $n = 1000$ and $d = 50$ and initialize $x = 0$. We sample minibatches of size $\tau = 10$ with replacement and use $\gamma = 0.1/\sqrt{LAK}$, where $K = 5000$ is the number of iterations and $A$ is as in Proposition 3. Similar to Needell & Ward (2017), we illustrate the utility of importance sampling by sampling $a_i$ from a zero-mean Gaussian of variance $i$ without normalizing the $a_i$. Since $L_i \propto \|a_i\|^2$, we can expect importance sampling to outperform uniform sampling in this case. However, when we normalize, the two methods should not be very different, and Figure 2 (of a single evaluation run) shows this.

### 7.2 Logistic regression with a nonconvex regularizer

We now consider the regularized logistic regression problem from (Tran-Dinh et al., 2019) with the aim of testing the fit of our Assumption 2 compared to other assumptions. The problem has the same form as (24) but with the logistic loss $l_i(x) = \log(1 + \exp(-a_i^T x))$ for $a_1, \ldots, a_n \in \mathbb{R}^d$ given and $\lambda = 0.5$. We run experiments on the a9a dataset ($n = 32561$ and $d = 123$) from LIBSVM (Chang & Lin, 2011). We fix $\tau = 1$, and run SGD for $K = 500$ iterations with a stepsize $\gamma = 1/\sqrt{LAK}$ as in the previous experiment. We use uniform sampling with replacement and measure the average squared stochastic gradient norm $1/n \sum_{i=1}^{n} \|\nabla f_i(x_k)\|^2$ every five iterations, in addition to the loss and the squared gradient norm. We then run nonnegative linear

least squares regression to fit the data for expected smoothness (ES) and compare to relaxed growth (RG). We also compare with theoretically estimated constants for (ES). The result is in Table 1, where we see a tight fit between our theory and the observed. The experimental setup and estimation details are explained more thoroughly in Section G.1 in the supplementary material.

### Acknowledgments

Ahmed Khaled acknowledges internship support from the Optimization and Machine Learning Lab led by Peter Richtárik at KAUST.

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

# Contents

## A Basic Facts and Notation

We will use the following facts from probability theory: If $X$ is a random variable and $Y$ is a constant vector, then

$$\mathbb{E}\left[\|Y - X\|^2\right] = \|Y - \mathbb{E}[X]\|^2 + \mathbb{E}\left[\|X - \mathbb{E}[X]\|^2\right]. \tag{25}$$

$$\mathbb{E}\left[\|X - \mathbb{E}[X]\|^2\right] = \mathbb{E}\left[\|X\|^2\right] - \|\mathbb{E}[X]\|^2. \tag{26}$$

A consequence of (26) is the following,

$$\mathbb{E}\left[\|X - \mathbb{E}[X]\|^2\right] \leq \mathbb{E}\left[\|X\|^2\right] \tag{27}$$

We will also make use of the following facts from linear algebra: for any $a, b \in \mathbb{R}^d$ and any $\zeta > 0$,

$$2\langle a, b \rangle = \|a\|^2 + \|b\|^2 - \|a - b\|^2. \tag{28}$$

$$\|a\|^2 \leq (1 + \zeta)\|a - b\|^2 + \left(1 + \zeta^{-1}\right)\|b\|^2. \tag{29}$$

For vectors $X_1, X_2, \ldots, X_n$ all in $\mathbb{R}^d$, the convexity of the squared norm $\|\cdot\|^2$ and a trivial application of Jensen's inequality yields the following inequality:

$$\left\|\frac{1}{n}\sum_{i=1}^n X_i\right\|^2 \leq \frac{1}{n}\sum_{i=1}^n \|X_i\|^2. \tag{30}$$

For an $L$-smooth function $f$ we have that for all $x, y \in \mathbb{R}^d$,

$$f(x) \leq f(y) + \langle \nabla f(y), x - y \rangle + \frac{L}{2}\|x - y\|^2. \tag{31}$$

$$\|\nabla f(x) - \nabla f(y)\| \leq L\|x - y\|. \tag{32}$$

Given a point $x \in \mathbb{R}^d$ and a stepsize $\gamma > 0$, we define the one step gradient descent mapping as,

$$T_\gamma(x) \stackrel{\text{def}}{=} x - \gamma \nabla f(x). \tag{33}$$

# B  Relations Between Assumptions

## B.1  Proof of Proposition 1

*Proof.* We define the function $f : \mathbb{R} \to \mathbb{R}$ as,

$$f(x) = \begin{cases} \frac{x^2}{2} & \text{if } |x| < 1, \\ |x| - \frac{1}{2} & \text{otherwise.} \end{cases}$$

Then $f$ is 1-smooth and lower bounded by 0. We consider using SGD with the stochastic gradient vector

$$g(x) = \begin{cases} \nabla f(x) + \sqrt{|x|} & \text{with probability } 1/2, \\ \nabla f(x) - \sqrt{|x|} & \text{with probability } 1/2. \end{cases}$$

Then suppose that (RG) holds, then there exists constants $\alpha$ and $\beta$ such that,

$$\mathbb{E}\left[\|g(x)\|^2\right] \leq \alpha\|\nabla f(x)\|^2 + \beta.$$

Consider $x = 2\max\{1, 2(\alpha + \beta)\}$, then $|x| > 1$ and hence $\nabla f(x) = 1$ by the definition of $f$. Specializing (RG) we get,

$$\mathbb{E}\left[\|g(x)\|^2\right] \leq \alpha + \beta. \tag{34}$$

On the other hand,

$$\begin{aligned}
\mathbb{E}\left[\|g(x)\|^2\right] = \frac{1}{2}\left(\left(1 + \sqrt{x}\right)^2 + \left(1 - \sqrt{x}\right)^2\right) &\geq \frac{1}{2}\left(1 + \sqrt{x}\right)^2 \\
&\geq \frac{1}{2}|x| = \max\{1, 2(\alpha + \beta)\}. \tag{35}
\end{aligned}$$

We see that (34) and (35) are in clear contradiction. It follows that (RG) does not hold. We now show that (ES) holds: first, suppose that $|x| \geq 1$, then

$$\begin{aligned}
\mathbb{E}\left[\|g(x)\|^2\right] &= \frac{1}{2}\left(\left(1 + \sqrt{|x|}\right)^2 + \left(1 - \sqrt{|x|}\right)^2\right) \\
&= \frac{1}{2}(2 + 2|x|) = 1 + |x|, \\
&= \frac{3}{2} + \left(f(x) - f^{\text{inf}}\right), \tag{36}
\end{aligned}$$

where in the last line we used that when $|x| \geq 1$ we have $f(x) - f^{\text{inf}} = |x| - 1/2$. Now suppose that $|x| \leq 1$, then

$$\begin{aligned}
\mathbb{E}\left[\|g(x)\|^2\right] &= \frac{1}{2}\left(\left(x + \sqrt{|x|}\right)^2 + \left(x - \sqrt{|x|}\right)^2\right) \\
&= x^2 + |x| \leq 1 + 1 = 2. \tag{37}
\end{aligned}$$

Combining (36) and (37) then we have that for all $x \in \mathbb{R}$,

$$\mathbb{E}\left[\|g(x)\|^2\right] \leq f(x) - f^{\text{inf}} + 2.$$

It follows that (ES) is satisfied with $A = 1/2$, $B = 0$, and $C = 2$. ∎

### B.2 Formal Statement and Proof of Theorem 1

**Theorem 1** (Formal) Suppose that $f : \mathbb{R}^d \to \mathbb{R}$ is $L$-smooth. Then the following relations hold:

1. The maximal strong growth condition (M-SG) implies the strong growth condition (E-SG).

2. The strong growth condition (E-SG) implies the relaxed growth condition (RG).

3. Bounded stochastic gradient variance (BV) implies the relaxed growth condition (RG).

4. The gradient confusion bound (GC) for finite-sum problems $f = \sum_{i=1}^n f_i / n$ implies the relaxed growth condition (RG).

5. The relaxed growth condition implies the expected smoothness condition (ES).

6. If $f$ has the expectation structure (7), then the sure-smoothness assumption implies the expected smoothness condition (ES).

*Proof.*      1. Suppose that (M-SG) holds. Then,

$$\mathbb{E}\left[\|g(x)\|^2\right] \leq \mathbb{E}\left[\alpha\|\nabla f(x)\|^2\right] = \alpha\|\nabla f(x)\|^2.$$

Hence (E-SG) holds.

2. If (E-SG) holds, then plugging $\beta = 0$ shows that (RG) holds.

3. If (BV) holds, then plugging $\alpha = 0$ shows that (RG) holds.

4. We use the definition of $\|\nabla f(x)\|^2$,

$$
\begin{aligned}
\|\nabla f(x)\|^2 &= \left\langle \sum_{i=1}^n \frac{\nabla f_i(x)}{n}, \sum_{j=1}^n \frac{\nabla f_j(x)}{n} \right\rangle \\
&= \frac{1}{n^2} \sum_{i=1}^n \sum_{j=1}^n \langle \nabla f_i(x), \nabla f_j(x) \rangle \\
&= \frac{1}{n^2} \left( \sum_{i=1}^n \|\nabla f_i(x)\|^2 + \sum_{i=1}^n \sum_{j=1, j\neq i}^n \langle \nabla f_i(x), \nabla f_j(x) \rangle \right) \\
&\geq \frac{1}{n} \left( \mathbb{E}_i\left[\|\nabla f_i(x)\|^2\right] - \frac{\eta(n^2 - n)}{n} \right) \\
&= \frac{1}{n} \mathbb{E}_i\left[\|\nabla f_i(x)\|^2\right] - \eta \cdot \left(1 - \frac{1}{n}\right).
\end{aligned}
$$

Rearranging we get,

$$\mathbb{E}\left[\|\nabla f_i(x)\|^2\right] \leq n \cdot \|\nabla f(x)\|^2 + \eta (n - 1).$$

Hence (RG) holds with $\alpha = n$ and $\beta = \eta (n - 1)$.

5. Putting $A = 0$, $B = \alpha$ and $C = \beta$ shows that (ES) is satisfied.

6. Note that $f_\xi(x)$ is almost surely bounded from below by 0, and that by (SS), $f_\xi(x)$ is almost-surely $L$-smooth. Then using Lemma 1 we have almost surely,

$$\|\nabla f_\xi(x)\|^2 \leq 2L f_\xi(x).$$

Let $f^{\text{inf}}$ be the infimum of $f$ (necessarily exists, since $f(x) = \mathbb{E}[f_\xi(x)] \geq 0$). Then,

$$\|\nabla f_\xi(x)\|^2 \leq 2L \left(f_\xi(x) - f^{\text{inf}}\right) + 2L f^{\text{inf}}.$$

Taking expectations and using that $f(x) = \mathbb{E}_{\xi \sim \mathcal{D}}\left[f_\xi(x)\right]$ we see that (ES) is satisfied with $A = 2L$ and $C = 2Lf^{\mathrm{inf}}$.

$\blacksquare$

## C   Proofs for Section 4.5 and 4.6

### C.1   Proof of Lemma 1

*Proof.* If we let $x_+ = x - \frac{1}{L}\nabla f(x)$, then using the $L$-smoothness of $f$ we get

$$f(x_+) \leq f(x) + \langle \nabla f(x), x_+ - x \rangle + \frac{L}{2}\|x_+ - x\|^2.$$

Using that $f^{\mathrm{inf}} \leq f(x_+)$ and the definition of $x_+$ we have,

$$f^{\mathrm{inf}} \leq f(x_+) \leq f(x) - \frac{1}{L}\|\nabla f(x)\|^2 + \frac{1}{2L}\|\nabla f(x)\|^2 = f(x) - \frac{1}{2L}\|\nabla f(x)\|^2.$$

Rearranging we get the claim. $\blacksquare$

### C.2   Proof of Proposition 2

*Proof.* We start out with the definition of $\nabla f_v(\cdot)$ then use the convexity of the squared norm $\|\cdot\|^2$ and the linearity of expectation:

$$
\begin{aligned}
\mathbb{E}\left[\|\nabla f_v(x)\|^2\right] &= \mathbb{E}\left[\left\|\frac{1}{n}\sum_{i=1}^n v_i \nabla f_i(x)\right\|^2\right] \\
&\overset{(30)}{\leq} \frac{1}{n}\sum_{i=1}^n \mathbb{E}\left[\|v_i \nabla f_i(x)\|^2\right] = \frac{1}{n}\sum_{i=1}^n \mathbb{E}\left[v_i^2\right]\|\nabla f_i(x)\|^2.
\end{aligned}
$$

We now use Lemma 1 and the nonnegativity of $f_i(x) - f_i^{\mathrm{inf}}$ (since $f_i^{\mathrm{inf}}$ is a lower bound on $f_i$),

$$
\begin{aligned}
\mathbb{E}\left[\|\nabla f_v(x)\|^2\right] &\overset{(1)}{\leq} \frac{2}{n}\sum_{i=1}^n L_i \mathbb{E}\left[v_i^2\right]\left(f_i(x) - f_i^{\mathrm{inf}}\right) \\
&\leq \frac{2\max_i\left(L_i \mathbb{E}\left[v_i^2\right]\right)}{n}\sum_{i=1}^n \left(f_i(x) - f_i^{\mathrm{inf}}\right) \\
&= \frac{2\max_i\left(L_i \mathbb{E}\left[v_i^2\right]\right)}{n}\sum_{i=1}^n \left(f_i(x) - f^{\mathrm{inf}} + f^{\mathrm{inf}} - f_i^{\mathrm{inf}}\right) \\
&= 2\max_i\left(L_i \mathbb{E}\left[v_i^2\right]\right)\left(f(x) - f^{\mathrm{inf}}\right) + 2\max_i\left(L_i \mathbb{E}\left[v_i^2\right]\right)\frac{1}{n}\sum_{i=1}^n \left(f^{\mathrm{inf}} - f_i^{\mathrm{inf}}\right).
\end{aligned}
$$

It remains to notice that since $f(x) = \frac{1}{n}\sum_{i=1}^n f_i(x) \geq \frac{1}{n}\sum_{i=1}^n f_i^{\mathrm{inf}}$, then $\frac{1}{n}\sum_{i=1}^n f_i^{\mathrm{inf}}$ is a lower bound on $f$, and hence

$$\Delta^{\mathrm{inf}} = \frac{1}{n}\sum_{i=1}^n \left(f^{\mathrm{inf}} - f_i^{\mathrm{inf}}\right) = f^{\mathrm{inf}} - \frac{1}{n}\sum_{i=1}^n f_i^{\mathrm{inf}} \geq 0,$$

as $f^{\mathrm{inf}}$ is the infimum (greatest lower bound) of $\{f(x) \mid x \in \mathbb{R}^d\}$ by definition. $\blacksquare$

### C.3 Proof of Proposition 3

*Proof.* From the definition of $\nabla f_v$ and the linearity of expectation,

$$
\begin{aligned}
\mathbb{E}\left[\|\nabla f_v(x)\|^2\right] &= \mathbb{E}\left[\left\|\frac{1}{n}\sum_{i=1}^n v_i\nabla f_i(x)\right\|^2\right] = \mathbb{E}\left[\left\langle\frac{1}{n}\sum_{i=1}^n v_i\nabla f_i(x),\frac{1}{n}\sum_{j=1}^n v_j\nabla f_j(x)\right\rangle\right] \\
&= \frac{1}{n^2}\sum_{i=1}^n\sum_{j=1}^n\mathbb{E}\left[v_iv_j\right]\langle\nabla f_i(x),\nabla f_j(x)\rangle.
\end{aligned}
\tag{38}
$$

We now consider each of the specific samplings individually:

(i) Independent sampling with replacement: we write

$$
v_i = \frac{S_i}{\tau q_i} = \frac{1}{\tau q_i}\sum_{k=1}^\tau S_{i,k},
\tag{39}
$$

where $S_{i,k}$ is defined by

$$
S_{i,k} = \begin{cases} 1 & \text{if the } k\text{-th dice roll resulted in } i, \\ 0 & \text{otherwise,} \end{cases}
$$

and clearly $\mathbb{E}\left[S_{i,k}\right] = q_i$. Then,

$$
\mathbb{E}\left[v_iv_j\right] = \frac{1}{\tau^2 q_i q_j}\sum_{k=1}^\tau\sum_{t=1}^\tau\mathbb{E}\left[S_{i,k}S_{j,t}\right].
\tag{40}
$$

It is straightforward to see by the independence of the dice rolls that

$$
\mathbb{E}\left[S_{i,k}S_{j,t}\right] = \begin{cases} q_i q_j & \text{if } k\neq t, \\ q_i & \text{if } k=t, i=j, \\ 0 & \text{if } k=t, i\neq j. \end{cases}
$$

Using this in (40) yields

$$
\mathbb{E}\left[v_iv_j\right] = \begin{cases} 1-\frac{1}{\tau} & \text{if } i\neq j, \\ 1-\frac{1}{\tau}+\frac{1}{\tau q_i} & \text{otherwise.} \end{cases}
$$

Plugging the last equality into (38), we get

$$
\begin{aligned}
\mathbb{E}\left[\|\nabla f_v(x)\|^2\right] &= \left(1-\frac{1}{\tau}\right)\frac{1}{n^2}\sum_{i=1}^n\sum_{j=1,j\neq i}^n\langle\nabla f_i(x),\nabla f_j(x)\rangle + \frac{1}{n^2}\sum_{i=1}^n\left(1-\frac{1}{\tau}+\frac{1}{\tau q_i}\right)\|\nabla f_i(x)\|^2 \\
&= \left(1-\frac{1}{\tau}\right)\frac{1}{n^2}\sum_{i=1}^n\sum_{j=1}^n\langle\nabla f_i(x),\nabla f_j(x)\rangle + \frac{1}{n^2}\sum_{i=1}^n\frac{\|\nabla f_i(x)\|^2}{\tau q_i} \\
&= \left(1-\frac{1}{\tau}\right)\left\langle\sum_{i=1}^n\frac{\nabla f_i(x)}{n},\sum_{j=1}^n\frac{\nabla f_j(x)}{n}\right\rangle + \frac{1}{n^2}\sum_{i=1}^n\frac{\|\nabla f_i(x)\|^2}{\tau q_i} \\
&= \left(1-\frac{1}{\tau}\right)\|\nabla f(x)\|^2 + \frac{1}{n^2}\sum_{i=1}^n\frac{\|\nabla f_i(x)\|^2}{\tau q_i}.
\end{aligned}
\tag{41}
$$

Using Lemma 1 on each $f_i$,

$$
\begin{aligned}
\mathbb{E}\left[\|\nabla f_v(x)\|^2\right] &\overset{(14)}{\leq} \left(1-\frac{1}{\tau}\right)\|\nabla f(x)\|^2 + \frac{2}{n}\sum_{i=1}^n\frac{L_i}{n\tau q_i}\left(f_i(x)-f_i^{\text{inf}}\right) \\
&\leq \left(1-\frac{1}{\tau}\right)\|\nabla f(x)\|^2 + 2\left(\max_i\frac{L_i}{n\tau q_i}\right)\frac{1}{n}\sum_{i=1}^n\left(f_i(x)-f_i^{\text{inf}}\right).
\end{aligned}
$$

It remains to use that $\frac{1}{n}\sum_{i=1}^{n}\left(f_i(x) - f_i^{\text{inf}}\right) = f(x) - f^{\text{inf}} + \Delta^{\text{inf}}$, and the nonnegativity of $\Delta^{\text{inf}}$ was proved in Proposition 2.

(ii) Independent sampling without replacement: it is not difficult to see that,

$$\mathbb{E}\left[v_i v_j\right] = \begin{cases} 1 & \text{if } i \neq j, \\ \frac{1}{p_i} & \text{if } i = j. \end{cases}$$

Using this in (38),

$$
\begin{aligned}
\mathbb{E}\left[\|\nabla f_v(x)\|^2\right] &= \frac{1}{n^2}\sum_{i=1}^{n}\sum_{j=1, j\neq i}^{n}\langle \nabla f_i(x), \nabla f_j(x)\rangle + \frac{1}{n^2}\sum_{i=1}^{n}\frac{1}{p_i}\|\nabla f_i(x)\|^2 \\
&= \frac{1}{n^2}\sum_{i=1}^{n}\sum_{j=1}^{n}\langle \nabla f_i(x), \nabla f_j(x)\rangle + \frac{1}{n^2}\sum_{i=1}^{n}\left(\frac{1}{p_i} - 1\right)\|\nabla f(x)\|^2.
\end{aligned}
$$

Continuing similarly to the previous sampling (from (41) onward), we get the required claim.

(iii) $\tau$-nice sampling without replacement: it is not difficult to see by elementary combinatorics that

$$\text{Prob}\left(i \in S \text{ and } j \in S\right) = \begin{cases} \frac{\tau}{n} & \text{if } i = j, \\ \frac{\tau(\tau-1)}{n(n-1)} & \text{otherwise.} \end{cases}$$

From this we can easily compute $\mathbb{E}\left[v_i v_j\right]$. Substituting into (38) and proceeding as the previous two cases yields the proposition's claim.

$\blacksquare$

### C.4 Proof of Proposition 4

*Proof.* First, it is easy to see that $g(x_k)$ is an unbiased estimator using the tower property of expectation and Assumption 4:

$$
\begin{aligned}
\mathbb{E}\left[g(x_k)\right] &= \frac{1}{n}\sum_{i=1}^{n}\mathbb{E}\left[\mathcal{Q}_{i,k}(g_i(x_k))\right] = \frac{1}{n}\sum_{i=1}^{n}\mathbb{E}\left[\mathbb{E}\left[\mathcal{Q}_{i,k}(g_i(x_k)) \mid g_i(x_k)\right]\right] \\
&= \frac{1}{n}\sum_{i=1}^{n}\mathbb{E}\left[g_i(x_k)\right] = \frac{1}{n}\sum_{i=1}^{n}\nabla f_i(x_k) = \nabla f(x_k).
\end{aligned}
$$

Second, to show that Assumption 2 is satisfied, we have for expectation conditional on $g_1(x_k), g_2(x_k), \ldots, g_n(x_k)$:

$$
\begin{aligned}
\mathbb{E}\left[\|g(x_k)\|^2\right] &= \mathbb{E}\left[\left\|\frac{1}{n}\sum_{i=1}^{n}\mathcal{Q}_{i,k}(g_i(x_k))\right\|^2\right] \\
&\stackrel{(26)}{=} \mathbb{E}\left[\left\|\frac{1}{n}\sum_{i=1}^{n}\left(\mathcal{Q}_{i,k}(g_i(x_k)) - g_i(x_k)\right)\right\|^2\right] + \left\|\frac{1}{n}\sum_{i=1}^{n}g_i(x_k)\right\|^2.
\end{aligned}
$$

Since $\mathcal{Q}_{1,k}, \mathcal{Q}_{2,k}, \ldots, \mathcal{Q}_{n,k}$ are independent by definition, the variance decomposes:

$$
\begin{aligned}
\mathbb{E}\left[\|g(x_k)\|^2\right] &= \frac{1}{n^2}\sum_{i=1}^{n}\mathbb{E}\left[\|\mathcal{Q}_{i,k}(g_i(x_k)) - g_i(x_k)\|^2\right] + \left\|\frac{1}{n}\sum_{i=1}^{n}g_i(x_k)\right\|^2 \\
&\stackrel{(18)}{\leq} \frac{1}{n^2}\sum_{i=1}^{n}\omega_i\|g_i(x_k)\|^2 + \left\|\frac{1}{n}\sum_{i=1}^{n}g_i(x_k)\right\|^2.
\end{aligned}
\tag{42}
$$

We now take expectation with respect to the randomness in the $g_i(x_k)$. We consider the first term in (42), where once again the variance decomposes by the independence of $g_1(x_k), g_2(x_k), \ldots, g_n(x_k)$:

$$\mathbb{E}\left[\left\|\frac{1}{n}\sum_{i=1}^{n}g_i(x_k)\right\|^2\right] \stackrel{(26)}{=} \mathbb{E}\left[\left\|\frac{1}{n}\sum_{i=1}^{n}g_i(x_k) - \nabla f_i(x_k)\right\|^2\right] + \|\nabla f(x_k)\|^2$$

$$= \frac{1}{n^2}\sum_{i=1}^{n}\mathbb{E}\left[\|g_i(x_k) - \nabla f_i(x_k)\|^2\right] + \|\nabla f(x_k)\|^2$$

$$\stackrel{(27)}{\leq} \frac{1}{n^2}\sum_{i=1}^{n}\mathbb{E}\left[\|g_i(x_k)\|^2\right] + \|\nabla f(x_k)\|^2. \tag{43}$$

Combining (42) and (43),

$$\mathbb{E}\left[\|g(x_k)\|^2\right] \leq \frac{1}{n^2}\sum_{i=1}^{n}(1+\omega_i)\,\mathbb{E}\left[\|g_i(x_k)\|^2\right] + \|\nabla f(x_k)\|^2. \tag{44}$$

For the first term in (44), we have using Assumption 2 and then Lemma 1,

$$\frac{1}{n^2}\sum_{i=1}^{n}(1+\omega_i)\,\mathbb{E}\left[\|g_i(x_k)\|^2\right] \leq \frac{1}{n^2}\sum_{i=1}^{n}(1+\omega_i)\left(2A_i\left(f_i(x_k) - f_i^{\inf}\right) + B_i\|\nabla f_i(x_k)\|^2 + C_i\right)$$

$$\leq \frac{1}{n^2}\sum_{i=1}^{n}(1+\omega_i)\left(2\left(A_i + B_iL_i\right)\left(f_i(x_k) - f_i^{\inf}\right) + C_i\right)$$

$$\leq \frac{1}{n}\sum_{i=1}^{n}\left(2A\left(f_i(x_k) - f_i^{\inf}\right)\right) + \frac{1}{n^2}\sum_{i=1}^{n}(1+\omega_i)\,C_i$$

$$= 2A\left(f(x_k) - f^{\inf}\right) + C. \tag{45}$$

where $A \stackrel{\text{def}}{=} \max_i (1+\omega_i)\left(A_i + B_iL_i\right)/n$, and $C \stackrel{\text{def}}{=} 2A\Delta^{\inf} + \frac{1}{n^2}\sum_{i=1}^{n}(1+\omega_i)\,C_i$, and where $\Delta^{\inf}$ is defined as in Proposition 2. Combining (45) with (44) we finally get,

$$\mathbb{E}\left[\|g(x_k)\|^2\right] \leq 2A\left(f(x_k) - f^{\inf}\right) + \|\nabla f(x_k)\|^2 + C,$$

which shows that Assumption 2 is satisfied. ∎

## D   Proofs for General Smooth Objectives

### D.1   Proof of Lemma 2

*Proof.* We start with the $L$-smoothness of $f$, which implies

$$f(x_{k+1}) \leq f(x_k) + \langle \nabla f(x_k), x_{k+1} - x_k \rangle + \frac{L}{2}\|x_{k+1} - x_k\|^2$$

$$= f(x_k) - \gamma\langle \nabla f(x_k), g(x_k)\rangle + \frac{L\gamma^2}{2}\|g(x_k)\|^2. \tag{46}$$

Taking expectations in (46) conditional on $x_k$, and using Assumption 2, we get

$$\mathbb{E}\left[f(x_{k+1}) \mid x_k\right] = f(x_k) - \gamma\|\nabla f(x_k)\|^2 + \frac{L\gamma^2}{2}\mathbb{E}\left[\|g(x_k)\|^2\right]$$

$$\stackrel{(\mathrm{ES})}{\leq} f(x_k) - \gamma\|\nabla f(x_k)\|^2 + \frac{L\gamma^2}{2}\left(2A\left(f(x_k) - f^{\inf}\right) + B\|\nabla f(x_k)\|^2 + C\right)$$

$$= f(x_k) - \gamma\left(1 - \frac{LB\gamma}{2}\right)\|\nabla f(x_k)\|^2 + L\gamma^2 A\left(f(x_k) - f^{\inf}\right) + \frac{L\gamma^2 C}{2}.$$

Subtracting $f^{\text{inf}}$ from both sides gives

$$\mathbb{E}\left[f(x_{k+1}) \mid x_k\right] - f^{\text{inf}} \leq \left(1 + L\gamma^2 A\right)\left(f(x_k) - f^{\text{inf}}\right) - \gamma\left(1 - \frac{LB\gamma}{2}\right)\|\nabla f(x_k)\|^2 + \frac{L\gamma^2 C}{2},$$

taking expectation again, using the tower property and rearranging, we get

$$\mathbb{E}\left[f(x_{k+1}) - f^{\text{inf}}\right] + \gamma\left(1 - \frac{LB\gamma}{2}\right)\mathbb{E}\left[\|\nabla f(x_k)\|^2\right] \leq \left(1 + L\gamma^2 A\right)\mathbb{E}\left[f(x_k) - f^{\text{inf}}\right] + \frac{L\gamma^2 C}{2}.$$

Letting $\delta_k \overset{\text{def}}{=} \mathbb{E}\left[f(x_k) - f^{\text{inf}}\right]$ and $r_k \overset{\text{def}}{=} \mathbb{E}\left[\|\nabla f(x_k)\|^2\right]$, we can rewrite the last inequality as

$$\gamma\left(1 - \frac{LB\gamma}{2}\right) r_k \leq \left(1 + L\gamma^2 A\right)\delta_k - \delta_{k+1} + \frac{L\gamma^2 C}{2}.$$

Our choice of stepsize guarantees that $1 - \frac{LB\gamma}{2} \geq \frac{1}{2}$. As such,

$$\frac{\gamma}{2} r_k \leq \left(1 + L\gamma^2 A\right)\delta_k - \delta_{k+1} + \frac{L\gamma^2 C}{2}. \tag{47}$$

We now follow Stich (2019) and define a *weighting sequence* $w_0, w_1, w_2, \ldots, w_K$ that is used to weight the terms in (47). However, unlike Stich (2019), we are interested in the weighting sequence solely as a proof technique, and it does not show up in the final bounds. Fix $w_{-1} > 0$. Define $w_k = \frac{w_{k-1}}{1+L\gamma^2 A}$ for all $k \geq 0$. Multiplying (47) by $w_k/\gamma$,

$$\frac{1}{2} w_k r_k \leq \frac{w_k\left(1 + L\gamma^2 A\right)}{\gamma}\delta_k - \frac{w_k}{\gamma}\delta_{k+1} + \frac{LCw_k\gamma}{2}$$

$$= \frac{w_{k-1}}{\gamma}\delta_k - \frac{w_k}{\gamma}\delta_{k+1} + \frac{LCw_k\gamma}{2}.$$

Summing up both sides as $k = 0, 1, \ldots, K-1$ we have,

$$\frac{1}{2}\sum_{k=0}^{K-1} w_k r_k \leq \frac{w_{-1}}{\gamma}\delta_0 - \frac{w_{K-1}}{\gamma}\delta_K + \frac{LC}{2}\sum_{k=0}^{K-1} w_k\gamma.$$

Rearranging we get the lemma's statement. $\blacksquare$

### D.2 A descent lemma

**Lemma 4.** Under Assumption 1 we have for any $\gamma > 0$,

$$f(x) - f(T_\gamma(x)) \leq \frac{\gamma}{2}\left(3L\gamma + 2\right)\|\nabla f(x)\|^2, \tag{48}$$

where $T_\gamma(x) \overset{\text{def}}{=} x - \gamma\nabla f(x)$.

*Proof.* Using the $L$-smoothness of $f$,

$$
\begin{aligned}
f(x) - f(T_\gamma(x)) &\leq \langle\nabla f(T_\gamma(x)), x - T_\gamma(x)\rangle + \frac{L}{2}\|x - T_\gamma(x)\|^2 \\
&\overset{(33)}{=} \gamma\langle\nabla f(T_\gamma(x)), \nabla f(x)\rangle + \frac{L\gamma^2}{2}\|\nabla f(x)\|^2 \\
&\overset{(28)}{=} \frac{\gamma}{2}\left(\|\nabla f(T_\gamma(x))\|^2 + \|\nabla f(x)\|^2 - \|\nabla f(T_\gamma(x)) - \nabla f(x)\|^2\right) + \frac{L\gamma^2}{2}\|\nabla f(x)\|^2 \\
&\overset{(29)}{\leq} \frac{\gamma}{2}\left(\zeta\|\nabla f(T_\gamma(x)) - \nabla f(x)\|^2 + \left(2 + \zeta^{-1}\right)\|\nabla f(x)\|^2\right) + \frac{L\gamma^2}{2}\|\nabla f(x)\|^2 \\
&\overset{(32)}{\leq} \frac{\gamma}{2}\left(L^2\zeta\|T_\gamma(x) - x\|^2 + \left(2 + \zeta^{-1}\right)\|\nabla f(x)\|^2\right) + \frac{L\gamma^2}{2}\|\nabla f(x)\|^2 \\
&\overset{(33)}{=} \frac{\gamma}{2}\left(L^2\zeta\gamma^2 + \zeta^{-1} + 2 + L\gamma\right)\|\nabla f(x)\|^2.
\end{aligned}
$$

Put $\zeta = \frac{1}{L\gamma}$,

$$f(x) - f(T_\gamma(x)) \leq \frac{\gamma}{2}\left(3L\gamma + 2\right)\|\nabla f(x)\|^2.$$ ∎

### D.3 Proof of Theorem 2

*Proof.* We start with Lemma 2,

$$\frac{1}{2}\sum_{k=0}^{K-1} w_k r_k \leq \frac{1}{2}\sum_{k=0}^{K-1} w_k r_k + \frac{w_{K-1}}{\gamma}\delta_K \leq \frac{w_{-1}}{\gamma}\delta_0 + \frac{LC}{2}\sum_{k=0}^{K-1} w_k \gamma.$$

Let $W_K = \sum_{k=0}^{K-1} w_k$. Dividing both sides by $W_K$ we have,

$$\frac{1}{2}\min_{0\leq k\leq K-1} r_k \leq \frac{1}{W_K}\sum_{k=0}^{K-1} w_k r_k \leq \frac{w_{-1}}{W_K}\frac{\delta_0}{\gamma} + \frac{LC\gamma}{2}. \tag{49}$$

Note that,

$$W_K = \sum_{k=0}^{K-1} w_k \geq \sum_{k=0}^{K-1}\min_{0\leq i\leq K-1} w_i = Kw_{K-1} = \frac{Kw_{-1}}{\left(1 + L\gamma^2 A\right)^K}. \tag{50}$$

Using this in (49),

$$\frac{1}{2}\min_{0\leq k\leq K-1} r_k \leq \frac{\left(1 + L\gamma^2 A\right)^K}{\gamma K}\delta_0 + \frac{LC\gamma}{2}.$$

Dividing both sides by $1/2$ yields the theorem's claim. ∎

### D.4 Proof of Corollary 1

*Proof.* From Theorem 2 under the condition that $\gamma \leq 1/(BL)$,

$$\min_{0\leq k\leq K-1}\mathbb{E}\left[\|\nabla f(x_k)\|^2\right] \leq LC\gamma + \frac{2\left(1 + L\gamma^2 A\right)^K}{\gamma K}\left(f(x_0) - f^{\inf}\right). \tag{51}$$

Using the fact that $1 + x \leq \exp(x)$, we have that

$$\left(1 + L\gamma^2 A\right)^K \leq \exp(L\gamma^2 AK) \leq \exp(1) \leq 3,$$

where the second inequality holds because $\gamma \leq 1/\sqrt{LAK}$ by assumption. Substituting in (51) we get,

$$\min_{0\leq k\leq K-1}\mathbb{E}\left[\|\nabla f(x_k)\|^2\right] \leq LC\gamma + \frac{6}{\gamma K}\left(f(x_0) - f^{\inf}\right). \tag{52}$$

To make the right hand side of (52) smaller than $\varepsilon^2$, we require that the first term satisfies

$$LC\gamma \leq \frac{\varepsilon^2}{2} \Rightarrow \gamma \leq \frac{\varepsilon^2}{2LC}.$$

Similarly for the second term, we get that the number of iterations must satisfy:

$$\frac{6\delta_0}{\gamma K} \leq \frac{\varepsilon^2}{2} \Rightarrow K \geq \frac{12\delta_0}{\gamma\varepsilon^2}. \tag{53}$$

Note that we have three requirements on the stepsize $\gamma$ so far:

$$\gamma \leq \frac{1}{BL} \qquad\qquad \gamma \leq \frac{\varepsilon^2}{2LC} \qquad\qquad \gamma \leq \frac{1}{\sqrt{LAK}}.$$

Plugging each of the previous bounds on the stepsize into (53),

$$K \geq \frac{12\delta_0 BL}{\varepsilon^2} \qquad\qquad K \geq \frac{24\delta_0 LC}{\varepsilon^4} \qquad\qquad K \geq \frac{12\delta_0\sqrt{LAK}}{\varepsilon^2}. \tag{54}$$

Since $K$ appears in both sides of the last requirement, by cancellation and squaring we simplify (54) to

$$K \geq \frac{12\delta_0 BL}{\varepsilon^2} \qquad\qquad K \geq \frac{24\delta_0 LC}{\varepsilon^4} \qquad\qquad K \geq \frac{12^2\delta_0^2 LA}{\varepsilon^4}.$$

Finally, we collect the terms into a single bound:

$$K \geq \frac{12\delta_0 L}{\varepsilon^2} \max\left\{B, \frac{2C}{\varepsilon^2}, \frac{12\delta_0 A}{\varepsilon^2}\right\}. \qquad\qquad \blacksquare$$

## E  Proofs Under Assumption 5

### E.1  Proof of Lemma 3

*Proof.* This proof loosely follows Lemma 3 in (Stich, 2019) without using the $(s_t)_t$ sequence as it is not relevant to our bounds and without using exponential averaging. If $K \leq \frac{b}{a}$, then starting from (20) we have for $\gamma = \frac{1}{b}$,

$$r_K \leq (1 - a\gamma)\, r_{K-1} + \gamma^2 c.$$

Recursing the above inequality we get,

$$
\begin{aligned}
r_K &\leq (1 - a\gamma)^K r_0 + \gamma^2 c \sum_{i=0}^{K-1} (1 - a\gamma)^i \\
&\leq (1 - a\gamma)^K r_0 + \frac{\gamma c}{a} \\
&\leq \exp(-a\gamma K)\, r_0 + \frac{\gamma c}{a} \\
&= \exp\left(-\frac{aK}{b}\right) r_0 + \frac{c}{ab} \\
&\leq \exp\left(-\frac{aK}{b}\right) r_0 + \frac{c}{a^2 K}. 
\end{aligned}
\tag{55}
$$

where in the last line we used that $K \leq \frac{b}{a}$. If $K \geq \frac{b}{a}$, then first we have

$$r_{k_0} \leq \exp\left(-\frac{aK}{2b}\right) r_0 + \frac{c}{ab}. \tag{56}$$

Then for $t > k_0$,

$$r_t \leq (1 - a\gamma_t)\, r_{t-1} + c\gamma_t^2 = \frac{s + t - k_0 - 2}{s + t - k_0} r_{t-1} + \frac{4c}{a^2 (s + t - k_0)^2}.$$

Multiplying both sides by $(s + t - k_0)^2$,

$$
\begin{aligned}
(s + t - k_0)^2 r_t &\leq (s + t - k_0 - 2)(s + t - k_0)\, r_{t-1} + \frac{4c}{a^2} \\
&= \left((s + t - k_0 - 1)^2 - 1\right) r_{t-1} + \frac{4c}{a^2} \\
&\leq (s + t - k_0 - 1)^2 r_{t-1} + \frac{4c}{a^2}.
\end{aligned}
$$

Let $w_t = (s + t - k_0)^2$. Then,

$$w_t r_t \quad \leq \quad w_{t-1} r_{t-1} + \frac{4c}{a^2}.$$

Summing up for $t = k_0 + 1, k_0 + 2, \ldots, K$ and telescoping we get,

$$w_K r_K \quad \leq \quad w_{k_0} r_{k_0} + \frac{4c\,(K - k_0)}{a^2} \quad = \quad s^2 r_{k_0} + \frac{4c\,(K - k_0)}{a^2}.$$

Dividing both sides by $w_K$ and using that since $s \geq k_0$ then $w_K = (s + K - k_0)^2 \geq (K - k_0)^2$,

$$r_K \quad \leq \quad \frac{s^2 r_{k_0}}{w_K} + \frac{4c(K - k_0)}{a^2 w_K} \quad \leq \quad \frac{s^2 r_{k_0}}{(K - k_0)^2} + \frac{4c}{a^2(K - k_0)}. \tag{57}$$

By the definition of $k_0$ we have $K - k_0 \geq K/2$. Plugging this estimate into (57)

$$r_K \quad \leq \quad \frac{4s^2 r_{k_0}}{K^2} + \frac{8c}{a^2 K} \tag{58}$$

Using (56) and the fact that $K^2 \geq \frac{b^2}{a^2} = 4s^2$ we have,

$$\frac{4s^2 r_{k_0}}{K^2} \quad \leq \quad \frac{4s^2}{K^2} \left( \exp\left( -\frac{aK}{2b} \right) r_0 + \frac{c}{ab} \right) \tag{59}$$

$$= \quad \frac{4s^2}{K^2} \exp\left( -\frac{aK}{2b} \right) r_0 + \frac{4cs^2}{abK^2} \tag{60}$$

$$\leq \quad \exp\left( -\frac{aK}{2b} \right) r_0 + \frac{4cs^2}{abK^2} \tag{61}$$

For the second term in (61), note that since $K \geq b/a$

$$\frac{4s^2}{bK} = \frac{4s^2}{b} \frac{1}{K} \leq \frac{4s^2}{b} \frac{a}{b} = \frac{1}{a}. \tag{62}$$

We substitute this into (61),

$$\frac{4s^2 r_{k_0}}{K^2} \quad \leq \quad \exp\left( -\frac{aK}{2b} \right) r_0 + \frac{c}{a^2 K}. \tag{63}$$

Substituting with (63) in (58),

$$r_K \quad \leq \quad \exp\left( -\frac{aK}{2b} \right) r_0 + \frac{9c}{a^2 K}. \tag{64}$$

It remains to take the maximum of the two bounds (64) and (55). ∎

### E.2 Proof of Theorem 3

*Proof.* Using the $L$-smoothness of $f$,

$$f(x_{k+1}) - f^\star \quad \leq \quad f(x_k) - f^\star + \langle \nabla f(x_t), x_{k+1} - x_k \rangle + \frac{L}{2} \|x_{k+1} - x_k\|^2$$

$$= \quad f(x_k) - f^\star - \gamma_k \langle \nabla f(x_k), g(x_k) \rangle + \frac{L\gamma_k^2}{2} \|g(x_k)\|^2.$$

Taking expectation conditional on $x_k$ and using Assumption 2,

$$\mathbb{E}\left[ f(x_{k+1}) - f^\star \right] \quad \leq \quad f(x_k) - f^\star - \gamma_k \|\nabla f(x_k)\|^2 + \frac{L\gamma_k^2}{2} \mathbb{E}\left[ \|g(x_k)\|^2 \right]$$

$$\overset{(\text{ES})}{\leq} \quad f(x_k) - f^\star - \gamma_k \left( 1 - \frac{LB\gamma_k}{2} \right) \|\nabla f(x_k)\|^2 + L\gamma_k^2 A \left( f(x_k) - f^\star \right) + \frac{L\gamma_k^2 C}{2}.$$

Using $1 - \frac{LB\gamma_k}{2} \geq \frac{3}{4}$ and Assumption 5,

$$\mathbb{E}\left[f(x_{k+1}) - f^\star\right] \leq \left(1 - \frac{3\gamma_k\mu}{2} + L\gamma_k^2 A\right)(f(x_k) - f^\star) + \frac{L\gamma_k^2 C}{2}.$$

Our choice of stepsize implies $L\gamma_k A \leq \frac{\mu}{2}$, hence

$$\mathbb{E}\left[f(x_{k+1} - f^\star)\right] \leq (1 - \gamma_k\mu)(f(x_k) - f^\star) + \frac{L\gamma_k^2 C}{2}.$$

Taking unconditional expectations and letting $r_k = \mathbb{E}\left[f(x_k) - f^\star\right]$ we have,

$$r_{k+1} \leq (1 - \gamma_k\mu) r_k + \frac{L\gamma_k^2 C}{2}.$$

Applying Lemma 3 with $a = \mu$, $b = \max\{2\kappa_f A, 2LB\}$, and $c = LC/2$ yields

$$\mathbb{E}\left[f(x_K) - f^\star\right] \leq \exp\left(-\frac{\mu K}{\max\{2\kappa_f A, 2LB\}}\right)(f(x_0) - f^\star) + \frac{9LC}{2\mu^2 K}.$$

Using the definition of $\kappa_S$ yields the theorem's statement. ∎

# F   Proofs Under Assumption 6

## F.1   A Lemma for Full Gradient Descent

**Lemma 5.** Under Assumption 6 and for $T_\gamma(x) = x - \gamma\nabla f(x)$ for $\gamma \leq \frac{1}{L}$ we have,

$$\|T_\gamma(x) - \pi(x)\|^2 \leq \|x - \pi(x)\|^2 - (1 - L\gamma)\gamma^2\|\nabla f(x)\|^2 - 2\gamma\left(f(T_\gamma(x)) - f^\star\right). \tag{65}$$

*Proof.* This is the second to last step in the proof of Theorem 12 in (Necoara et al., 2019), and we reproduce it for completeness. Let $y \in \mathbb{R}^d$. Then using the definition of $T_\gamma(x)$,

$$
\begin{aligned}
\|T_\gamma(x) - y\|^2 &= \|T_\gamma(x) - x + x - y\|^2 \tag{66}\\
&= \|x - y\|^2 + 2\langle x - y, T_\gamma(x) - x\rangle + \|T_\gamma(x) - x\|^2\\
&= \|x - y\|^2 + 2\langle T_\gamma(x) - y, T_\gamma(x) - x\rangle - \|T_\gamma(x) - x\|^2\\
&\overset{(33)}{=} \|x - y\|^2 - 2\gamma\langle\nabla f(x), T_\gamma(x) - y\rangle - \|T_\gamma(x) - x\|^2\\
&= \|x - y\|^2 - 2\gamma\left(\langle\nabla f(x), T_\gamma(x) - y\rangle + \frac{L}{2}\|T_\gamma(x) - x\|^2 + \left(\frac{1}{2\gamma} - \frac{L}{2}\right)\|T_\gamma(x) - x\|^2\right)\\
&= \|x - y\|^2 + (L\gamma - 1)\|T_\gamma(x) - x\|^2 - 2\gamma\left(\langle\nabla f(x), T_\gamma(x) - y\rangle + \frac{L}{2}\|T_\gamma(x) - x\|^2\right). \tag{67}
\end{aligned}
$$

We isolate the third term in (67),

$$\langle\nabla f(x), T_\gamma(x) - y\rangle + \frac{L}{2}\|T_\gamma(x) - x\|^2 = \langle\nabla f(x), x - y\rangle + \langle\nabla f(x), T_\gamma(x) - x\rangle + \frac{L}{2}\|T_\gamma(x) - x\|^2. \tag{68}$$

Since $f$ is $L$-smooth, then

$$f(T_\gamma(x)) - f(x) \leq \langle\nabla f(x), T_\gamma(x) - x\rangle + \frac{L}{2}\|T_\gamma(x) - x\|^2$$

Using this in (68) and using convexity,

$$
\begin{aligned}
\langle\nabla f(x), T_\gamma(x) - y\rangle + \frac{L}{2}\|T_\gamma(x) - x\|^2 &\geq \langle\nabla f(x), x - y\rangle + f(T_\gamma(x)) - f(x)\\
&\geq f(x) - f(y) + f(T_\gamma(x)) - f(x) = f(T_\gamma(x)) - f(y). \tag{69}
\end{aligned}
$$

Using (69) in (67),

$$\|T_\gamma(x) - y\|^2 \leq \|x - y\|^2 - (1 - L\gamma)\|T_\gamma(x) - x\|^2 - 2\gamma\left(f(T_\gamma(x)) - f(y)\right).$$

It remains to use that $T_\gamma(x) - x = -\gamma\nabla f(x)$ and to put $y = \pi(x)$. ∎

## F.2 Proof of Theorem 4

*Proof.* For expectation conditional on $x_k$, we have

$$
\begin{aligned}
\mathbb{E}\left[\|x_{k+1} - \pi(x_k)\|^2\right] &= \mathbb{E}\left[\|x_k - \pi(x_k) - \gamma_k g(x_k)\|^2\right] \\
&\overset{(25)}{=} \|x_k - \gamma_k \nabla f(x_k) - \pi(x_k)\|^2 + \gamma_k^2 \cdot \mathbb{E}\left[\|g(x_k) - \nabla f(x_k)\|^2\right] \\
&\overset{(26)}{=} \|x_k - \gamma_k \nabla f(x_k) - \pi(x_k)\|^2 + \gamma_k^2 \left(\mathbb{E}\left[\|g(x_k)\|^2\right] - \|\nabla f(x_k)\|^2\right) \\
&\overset{(65)}{\leq} \|x_k - \pi(x_k)\|^2 - (2 - L\gamma_k)\gamma_k^2 \|\nabla f(x_k)\|^2 - 2\gamma_k \left(f(T_{\gamma_k}(x_k)) - f^\star\right) \\
&\qquad + \gamma_k^2 \cdot \mathbb{E}\left[\|g(x_k)\|^2\right].
\end{aligned}
\tag{70}
$$

We can decompose the function decrease term and then use Lemma 4 as follows,

$$
\begin{aligned}
-2\gamma_k \left(f(T_{\gamma_k}(x_k)) - f^\star\right) &= -2\gamma_k \left(f(x_k) - f^\star\right) + 2\gamma_k \left(f(x_k) - f(T_{\gamma_k}(x_k))\right) \\
&\overset{(48)}{\leq} -2\gamma_k \left(f(x_k) - f^\star\right) + \gamma_k^2 \left(3L\gamma_k + 2\right)\|\nabla f(x_k)\|^2.
\end{aligned}
\tag{71}
$$

Let $r_k = \|x_k - \pi(x_k)\|^2$. Continuing from (70),

$$
\begin{aligned}
\mathbb{E}\left[\|x_{k+1} - \pi(x_k)\|^2\right] &\leq r_k - (2 - L\gamma_k)\gamma_k^2 \|\nabla f(x_k)\|^2 - 2\gamma_k \left(f(T_{\gamma_k}(x_k)) - f^\star\right) + \gamma_k^2 \cdot \mathbb{E}\left[\|g(x_k)\|^2\right] \\
&\overset{(71)}{\leq} r_k - 2\gamma_k \left(f(x_k) - f^\star\right) + 4L\gamma_k^3 \cdot \|\nabla f(x)\|^2 + \gamma_k^2 \cdot \mathbb{E}\left[\|g(x_k)\|^2\right].
\end{aligned}
\tag{72}
$$

Using Assumption 2 and Lemma 1 applied on $f$,

$$
\begin{aligned}
\mathbb{E}\left[\|g(x_k)\|^2\right] &\leq 2A\left(f(x_k) - f^\star\right) + B\|\nabla f(x_k)\|^2 + C \\
&\leq (2A + 2BL)\left(f(x_k) - f^\star\right) + C = 2\rho\left(f(x_k) - f^\star\right) + C,
\end{aligned}
\tag{73}
$$

where $\rho \overset{\text{def}}{=} A + BL$. We now use Lemma 1 to bound the squared gradient term:

$$
\begin{aligned}
\mathbb{E}\left[\|x_{k+1} - \pi(x_k)\|^2\right] &\overset{(72)}{\leq} r_k - 2\gamma_k \left(f(x_k) - f^\star\right) + 4L\gamma_k^3 \cdot \|\nabla f(x_k)\|^2 + \gamma_k^2 \cdot \mathbb{E}\left[\|g(x_k)\|^2\right] \\
&\overset{(14)}{\leq} r_k - 2\gamma_k \left(1 - 4L^2\gamma_k^2\right)\left(f(x_k) - f^\star\right) + \gamma_k^2 \cdot \mathbb{E}\left[\|g(x_k)\|^2\right] \\
&\overset{(73)}{\leq} r_k - 2\gamma_k \left(1 - 4L^2\gamma_k^2 - \gamma_k\rho\right)\left(f(x_k) - f^\star\right) + \gamma_k^2 C.
\end{aligned}
\tag{74}
$$

Now note that our choice of $\gamma_k$ guarantees that $1 - 4L^2\gamma_k^2 - \gamma_k\rho \geq \frac{1}{2}$, using this and Assumption 6 in (74) we get,

$$
\mathbb{E}\left[\|x_{k+1} - \pi(x_k)\|^2\right] \leq r_k - \gamma_k \left(f(x_k) - f^\star\right) + \gamma_k^2 C \leq \left(1 - \frac{\gamma_k\mu}{2}\right)r_k + \gamma_k^2 C.
$$

We now use the property of projections that $\mathbb{E}\left[\|x_{k+1} - \pi(x_{k+1})\|^2\right] \leq \mathbb{E}\left[\|x_{k+1} - y\|^2\right]$ for any $y \in X^\star$. Specializing this with $y = \pi(x_k)$ we get,

$$
\mathbb{E}\left[\|x_{k+1} - \pi(x_{k+1})\|^2\right] \leq \left(1 - \frac{\gamma_k\mu}{2}\right)\|x_k - \pi(x_k)\|^2 + \gamma_k^2 C.
$$

Taking unconditional expectation we get,

$$
\mathbb{E}\left[r_{k+1}\right] \leq \left(1 - \frac{\gamma_k\mu}{2}\right)\mathbb{E}\left[r_k\right] + \gamma_k^2 C.
$$

Using Lemma 3 with $a = \frac{\mu}{2}$, $b = \max\{4L, 4BL + 4A\}$, and $c = C$ we recover

$$
\mathbb{E}\left[r_K\right] \leq \exp\left(\frac{-\mu K}{2\max\{4L, 4BL + 4A\}}\right)r_0 + \frac{36C}{\mu^2 K}.
\tag{75}
$$

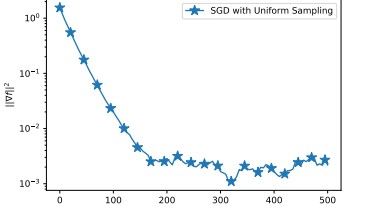 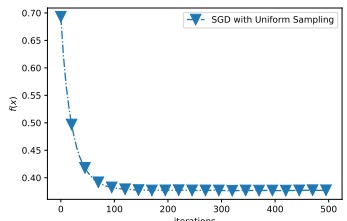 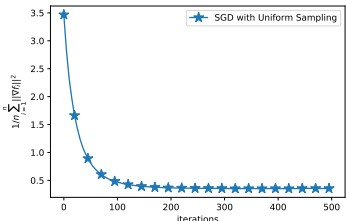

Figure 3: Results from SGD applied to the regularized logistic regression problem.

Equation (75) gives a convergence guarantee in terms of the distance of the iterates from the set of optima. We may convert this to convergence in function values using the following consequences of the quadratic functional growth property and smoothness: $r_0 \leq \frac{2}{\mu}(f(x_0) - f^\star)$ and $f(x_K) - f^\star \leq \frac{L}{2}r_K$. Hence,

$$\mathbb{E}\left[f(x_K) - f^\star\right] \leq \frac{1}{2}\kappa_f \exp\left(\frac{-\mu K}{2\max\{4L, 4BL + 4A\}}\right)(f(x_0) - f^\star) + \frac{18\kappa_f C}{\mu K}. \qquad \blacksquare$$

## G  Experimental Details

### G.1  Logistic regression with a nonconvex regularizer

**Experimental Setup**: The optimization problem considered is

$$\min_{x \in \mathbb{R}^d}\left\{\frac{1}{n}\sum_{i=1}^{n}\log\left(1 + \exp(-a_i^T x)\right) + \lambda\sum_{j=1}^{d}\frac{x_j^2}{1 + x_j^2},\right\}$$

where $a_1, a_2, \ldots, a_n \in \mathbb{R}^d$ are from the a9a dataset with $n = 32561$ and $d = 123$. We set $\lambda = 1/2$. We run SGD for $K = 500$ iterations with minibatch size 1 with uniform sampling.

**Data Collected**: We collect a snapshot once every five iterates (so $x_0, x_5, x_{15}, \ldots, x_{500}$) consisting of the squared full gradient norm $\|\nabla f(x_k)\|^2$, the loss on the entire dataset $f(x_k)$, and the average stochastic gradient norm $\mathbb{E}\left[\|g(x)\|^2\right] = 1/n\sum_{i=1}^{n}\|\nabla f_i(x_k)\|^2$. These are plotted in Figure 3. The full gradient norms go to zero, as expected, but the stochastic gradient norms decrease and quickly however around 0.5, and the full loss shows a similar pattern. We note that Lemma 2 can explain the plateau pattern, as using it one can show that provided the stepsize is set correctly, the loss at the last point does not exceed the initial loss in expectation.

**Hyperparameter Estimation**: We estimate the individual Lipschitz constants as $L_i \approx \frac{\|a_i\|^2}{4} + 2\lambda$. We estimate the global Lipschitz constant $L$ by their average $\bar{L} = \frac{1}{n}L_i$. We estimate $A$ for independent sampling as in Proposition 3 by $A = \max_i L_i/(\tau n q_i) = \max_i L_i$ as $\tau = 1$ and $q_i = 1/n$ for all $i$. We estimate $f^{\text{inf}}$ as the minimum value encountered over the entire SGD run and estimate $f_i^{\text{inf}}$ by running gradient descent for 200 iterations for each $f_i$ and reporting the minimum value encountered. The stepsize we use is then $\gamma = 1/\sqrt{LKA}$.

**Data Fitting**: For Assumption 2, the parameters $A, B, C \geq 0$ model the average stochastic gradient norms in terms of the loss and the full gradient. To find the best such parameters empirically, we solve the following optimization problem, minimizing the *squared residual error*

$$\min_{z \in \mathbb{R}^3, z \geq 0}\left\{g(z) = \|Wz - b\|^2\right\}, \tag{76}$$

where $z = \begin{bmatrix} z_1 & z_2 & z_3 \end{bmatrix}$ are the constants to be fit corresponding to $2A$, $B$, and $C$ respectively. $W$ is an $m \times 3$ matrix (for $m$ the number of snapshots, equal to 100 here) such that $W_{i,1} = f(x_i) - f^{\text{inf}}$, $W_{i,2} = \|\nabla f(x_i)\|^2$, and $W_{i,3} = 1$. Moreover, $b \in \mathbb{R}^m$ is the vector of average stochastic gradient norms, i.e. $b_k = 1/n\sum_{i=1}^{n}\|\nabla f_i(x_k)\|^2$.

Problem (76) is a nonnegative linear least squares problem and can be solved efficiently by linear algebra toolboxes. To model relaxed growth, we solve a similar problem but with only the parameters $B$ and $C$ (and with corresponding changes to the matrix $W$).

**Discussion of Table 1**: Table 1 shows that (ES) can achieve a smaller residual error compared to (RG) by relying on the function values to model the stochastic gradients. The value of the first coefficient is quite close to the theoretically estimated value of $L_{\max}$. The value of the coefficient $C$, however, is difficult to estimate. This is because according to our theory it depends on the *global* greatest lower bounds of the losses $f_i$ and $f$. In the case of convex objectives, however, this constant can be calculated well given knowledge of the minimizers of all the $f_i$ and of $f$.

## H  More on (Polyak & Tsypkin, 1973)

Polyak & Tsypkin (1973) study minimizing a function $f$ that is lower-bounded and whose gradient is $L$-Lipschitz. They study iterations of the form $x_n = x_{n-1} - \gamma_n s_n$, where $\gamma_n > 0$ is a stepsize and $s_n$ is a pseudogradient satisfying the following two inequalities:

$$0 \leq \langle \nabla f(x_{n-1}), \mathbb{E}[s_n] \rangle,$$
$$\mathbb{E}\left[ \|s_n\|^2 \right] \leq \lambda_n + K_1 f(x_{n-1}) + K_2 \langle \nabla f(x_{n-1}), \mathbb{E}[s_n] \rangle.$$

In addition, Polyak & Tsypkin (1973) assume the step sizes $\gamma_n$ and the sequence $\lambda_n$ jointly satisfy $\Sigma_{i=1}^{\infty} \lambda_n \gamma_n < \infty$ and $\Sigma_{n=1}^{\infty} \gamma_n = \infty$, and then additionally that either (a) $\Sigma_{i=1}^{\infty} \gamma_n^2 < \infty$ or (b) $\lambda_n = 0$, $K_1 = 0$, and $\lim_{n \to \infty} \gamma_n < \frac{2}{LK_2}$. Under these assumptions, they prove that $\lim_{n \to \infty} \langle \nabla f(x_{n-1}), \mathbb{E}[s_n] \rangle = 0$. In the stochastic setting, we have that $S_n = g(x_{n-1})$, and as such their assumptions reduce to:

(a) An assumption similar to (ES): $\mathbb{E}\left[ |g(x_{n-1})|^2 \right] \leq \lambda_n + K_1 f(x_{n-1}) + K_2 \|\nabla f(x_{n-1})\|^2$.

(b) $\Sigma_{i=1}^{\infty} \lambda_n \gamma_n < \infty$ and $\Sigma_{n=1}^{\infty} \gamma_n = \infty$.

(c) Either (1) $\Sigma_{i=1}^{\infty} \gamma_n^2 < \infty$ or (2) $\lambda_n = 0$, $K_1 = 0$, and $\lim_{n \to \infty} \gamma_n < \frac{2}{LK_2}$.

Part (a) would subsume our assumption if we could set $\lambda_n$ to be a constant, but the conditions on the sequences $\lambda_n$ and $\gamma_n$ in parts (b) and (c) are not the same as in our work– we explicitly set the stepsize instead, and obtain non-asymptotic results.

