# OpenReview forum: "Better Theory for SGD in the  Nonconvex World"
_TMLR — Accepted by TMLR_

### Review · Reviewer_kLwJ · 2022-10-16

**Summary Of Contributions:**

In this paper, authors proposed a new condition, expected smoothness (ES), for gradient estimator in stochastic nonconvex optimization. Authors claimed that the ES conditon is the most general one compared to most existing literature, and works for subsampling and compression strategies. With the new condtion, authors recovered $\mathcal{O}(\epsilon^{-4})$ complexity for general nonconvex problems, and $\mathcal{O}(\epsilon^{-1})$ complexity with an extra PL condtion.

**Broader Impact Concerns:**

This is a purely theory paper, I think there is no emergent ethical concerns.

**Requested Changes:**

Generally I hope authors can add more details or discussion to further justify the insights of the proposed assumption and the results. See above questions for more details.

**Strengths And Weaknesses:**

Strength:

1. The theory of the paper is relatively elegant, the flow of the paper is clear. The proof of the paper is kind of straightforward.
2. The theory provides a more general conditions compared to existing works, also it recovers existing $\mathcal{O}(\epsilon^{-4})$ complexity results.

Weakness:
1. I may be a bit concerned on whether the results is that insighful (I understand such judgement is a bit subjective, and TMLR tries to avoid 'significance/impact/novelty' in review). Personally I may view the proposed condition as an aggregation of the bounded variance condition (BV) and C-ES, so I may view the analysis is also an aggregation of the analysis of SGD on BV and SGD on C-ES (assuming the transfer from convex to nonconvex problems here is not that hard), so the convergence should be expected. I may highly appreciate if authors can provide more insights on the analysis compared to existing literature, e.g., what specifically the gap that authors find exciting and successfully get through.
2. With the new assumption, basically it requires users to know more problem parameters (e.g., Lipschitz constants of each $f_\xi$, rather than a unified $L_{\text{max}}$, as shown in the importance sampling example), which may not be that practical regarding the proposed more "practical" assumption (kind of trade-off?). With more information on the problem, getting a boost in the convergence with the importance sampling should be expected. So I hope to learn more about the importance and insight of the proposed new assumption.

Some other questions:
1. As a relaxation of the sure-smoothness (or each $f_\xi$ is Lipschitz smooth), does the averaged smooth condition ($\mathbb{E}_\xi||\nabla f_\xi(x)-\nabla f_\xi(y)||^2\leq L^2||x-y||^2$) in many literature (e.g., SPIDER) implies the ES condition?
2. In the subsampling and compression examples, the problem of interest is reduced to the form of finite-sum (rather than the purely stochastic form), many results come with a form of $\mathcal{O}(\text{poly}(n)\epsilon^{-2})$ (e.g., full GD will naturally attains an $\mathcal{O}(n\epsilon^{-2})$ complexity), rather than the $\mathcal{O}(\epsilon^{-4})$ here, which exhibits some advantages in terms of the $\epsilon$-dependence. Would you mind commenting on the difference between these two complexity results?

I appreciate authors' efforts on the work, please definitely indicate here if I misunderstand the details. Thank you very much.

---

> ### Author Response · Authors · 2022-11-04
> **Comments to Reviewer**
>
> Thank you so much for your review. Please find our responses below:
>
> 1) There are two main gaps in the literature that we hope to fill. The first is to provide a useful hierarchy of the current assumptions, and come up with something general and applicable that can be used to analyze the convergence of SGD. There are many assumptions in the literature, and we believe enumerating, comparing, and generalizing them is useful-- moreso that our results show the generalized assumption we give is strictly more general and captures settings not captured by existing assumptions. The second is to conduct a convergence analysis under this assumption, on this we have (a) a convergence result under smoothness & ES, (b) a convergence result under smoothness & ES, and the PL inequality, and (c) a convergence result when each the average function is convex, but the stochastic gradients can come from a nonconvex function. Our results on (a) and (b) are strict generalizations of prior work, and capture the settings that prior work did not capture. Moreover, our result on (c) improves upon prior work by providing a better convergence rate (showing dependence on $\kappa$ rather than $\kappa^2$).
>
> 2) We agree that knowing more parameters about the problem should naturally enable us to achieve better rates. However, the natural way of estimating $L_{\mathrm{max}}$ requires estimating $L_i$ for each $i$, or at least upper bounding each of them. Since estimating the smoothness parameter $L$ of the average function alone does not suffice anyway, we naturally have to estimate the smoothness parameter of each function in the finite-sum.
>
> 3) The mean-squared smoothness assumption alone does not imply ES, as an example take $f_1 (x) = x^2/2$ and $f_2 (x) = -x^2/2$ then $f = (f_1 + f_2)/2 = 0$ and ES can not hold since $E[\|g(x)\|^2] = \|x\|^2$ which cannot be bounded by a constant. However, $(f_1, f_2)$ satisfy mean-squared smoothness with constant $1$. However, this should come as no surprise: the mean-squared smoothness assumption often has to be paired with another structural assumption, such as bounded variance (in the online case, e.g. [1, Thm 3] or [2, Thm 1]) or finite-sum structure (as in SPIDER [2, Thm 2]).
>
> 4) The rate of GD can be captured by our framework by putting $A = 0$, $B = 1$, and $C = 0$. This immediately yields an $\mathcal{O} (\varepsilon^{-2})$ iteration complexity. Computing each of these gradients in a finite-sum requires $n$ individual function gradient evaluations, yielding the total sample complexity of $\mathcal{O} (n \varepsilon^{-2})$. Generally, the performance of SGD is better when the number of objectives $n$ is very large.
>
> [1] Yossi Arjevani, Yair Carmon, John C. Duchi, Dylan J. Foster, Nathan Srebro, Blake Woodworth. Lower Bounds for Non-Convex Stochastic Optimization, Mathematical Programming (2022).
> [2] Cong Fang, Chris Junchi Li, Zhouchen Lin, Tong Zhang. SPIDER: Near-Optimal Non-Convex Optimization via Stochastic Path-Integrated Differential Estimator, NeurIPS 2018.

---

### Review · Reviewer_VX3G · 2022-10-28

**Summary Of Contributions:**

This paper proposes an expected smoothness assumption in a general form that is looser than other assumptions in the literature and studies the convergence of SGD for nonconvex problems. Specifically, this work shows the convergence to a stationary point for smooth nonconvex problems and convergence to the optimal value with an additional PL-condition or quadratic growth condition with better rates. Moreover, several interesting examples that satisfy an expected smoothness condition are provided.

**Broader Impact Concerns:**

Not applicable.

**Requested Changes:**

- The discussion about the relationship with the ER condition would be helpful as commented above.


Minor comment:
- Typo: The last equation of the proof of Theorem 4 on page 31. The exponential is missing.

**Strengths And Weaknesses:**

**Strengths**.

- This paper is well-written and easy to read.
- The introduced expected smoothness condition is quite nice. Indeed, there are several important examples (e.g., subsampling and compression) that satisfy this condition but do not satisfy other assumptions used in
past works.

**Weaknesses**.

- The proofs are straightforward and not so novel, which are basically collections of the existing techniques. (But, I acknowledge the significance of the paper as commented in strength.)

- Recently, a closely relevant paper [Gower et al. (2021)] was published. They introduced an expected residual (ER) condition which is also an extension of the standard expected smoothness, although ER is a special case of the condition proposed in this submission.
My concern is the difference between these conditions. Is there any example that does not satisfy ER but satisfies the proposed expected smoothness condition?

[Gower et al. (2021)] Robert M. Gower, Othmane Sebbouh, and Nicolas Loizou. SGD for Structured Nonconvex Functions: Learning Rates, Minibatching and Interpolation. AISTATS, 2021.

---

> ### Author Response · Authors · 2022-11-17
> **Comments to Reviewer**
>
> We thank you so much for reviewing our paper. Please find our responses below:
>
> 1. (Novelty). While our proofs use a lot of existing techniques, we would like to bring attention to two theorems: Theorem 3 achieves the best-known convergence rate for vanilla SGD applied PL functions _without requiring knowledge of the noise parameter $\sigma$_, this is known as noise-adaptivity and is an important goal in its own, and the work of (Vaswani, Dubois-Taine, Babanezhad, Towards Noise-adaptive, Problem-adaptive (Accelerated) SGD, ICML 2022) mentions this-- at the time of writing our paper, this was also the first result to achieve this. Moreover, Theorem 4 also achieves a better convergence result compared to existing literature when the non-convexity is averaged out. Both of these are novel results in their own right.
>
> 2. (Discussion of Gower et al. (2021) and the ER condition). First, we note that the work of Gower et al. (2021) succeeds ours, and cites our work; We do not think the comparison with their work is fair against a paper that came out after ours and built upon the results we had. Second, it is not difficult to construct examples that satisfy ES but would yield suboptimal results under ER: In general, by (Gower et al., 2021, p. 4 eqn (7)) we see that ER cannot easily adapt when the constant in front of the gradient is larger than $1$. This holds when the gradient estimator satisfies strong growth: for example, let $g$ be any gradient estimator satisfying the strong growth condition with $\alpha > 1$, then $g(x_*) = \nabla f(x_*) = 0$ almost surely and $\mathbb{E} \| g(x) \|^2 <= \alpha \| \nabla f(x) \|^2$, then because $\| \nabla f(x) \|^2 <= 2L (f(x) - f_*) = 2 L (f(x) - f_*)$,  and ER is satisfied with $\rho = 2 L \alpha$. If $f$ additionally satisfies the PL inequality, then (Gower et al., 2021, Theorem 4.6) gives a sample complexity of $\tilde{\mathcal{O}} \left (  \frac{L}{\mu} \frac{L \alpha}{ \mu }  \right )$, but Corollary 2 in our work gives a sample complexity of $\tilde{\mathcal{O}} \left ( \frac{L}{\mu} \alpha \right )$, when $\frac{L}{\mu}$ is large, this difference can be very large.
>
> 3. Thank you for telling us about the typo, we will fix it.

---

### Review · Reviewer_1v7k · 2022-11-03

**Summary Of Contributions:**

This paper studies the convergence rate of SGD based on a general assumption of the objective function, that can cover multiple designs of stochastic gradients.

In particular, this paper proposes a new assumption that can be reduced to a set of existing assumptions.

Then the authors show that the assumption can cover the stochastic gradients that are designed based on a combination of gradient compression and subsampling, which cannot be described by prior assumptions.

Finally, the authors develop the convergence rate of SGD for general smooth nonconvex problems and the problems satisfying the PL condition, which recover the existing convergence results of SGD.

**Broader Impact Concerns:**

No concern.

**Requested Changes:**

1. Detailed comparison with prior work [Polyak & Tsypkin, 1973] should be added.
2. Convergence rate of stochastic variance reduced gradients should be considered.
3. The authors should reorganize the paper to pay more attention to the theoretical results.

**Strengths And Weaknesses:**

Strength:

* The proposed assumption can cover existing assumptions, and the convergence results can also be adapted accordingly.
* The proposed assumption can cover certain stochastic gradients that cannot be described by prior works.

Weakness:

* The novelty may be limited. As claimed by the authors in the text after Assumption 2, the expected smoothness has already been discovered in early works. Therefore, such a "rediscovery" cannot be viewed as a significant novelty. Besides, a detailed version of this assumption made in [Polyak & Tsypkin, 1973] should also be presented for comparison.

* The depth of this paper is also not enough. It has been widely known that variance-reduced stochastic gradients can significantly improve the convergence in both convex and nonconvex optimization problems. Then what can we get if the problem is characterized by such as new assumption proposed in this paper?

* The authors have paid too much attention to the background and introduction of existing assumptions and stochastic gradients, while the authors only present the key theoretical results within 3 pages.

---

> ### Author Response · Authors · 2022-11-18
> **Comments to reviewer**
>
> 1. (Comparison with Polyak & Tsypkin 1973). They study minimizing a function $f$ that is lower-bounded and whose gradient is $L$-Lipschitz. The iterations $x_n = x_{n-1} - \gamma_n s_n$, where $\gamma_n > 0$ is a stepsize and $s_n$ is a pseudogradient satisfying $\langle \nabla f (x_{n-1}), \mathbb{E}[ s_n ] \rangle \geq 0$. And, more importantly, $\mathbb{E} \left[ | s_n |^2 \right] \leq \lambda_n + K_1 f(x_{n-1}) + K_2 \langle ∇ f(x_{n-1}), \mathbb{E} [s_n] \rangle$. They assume the step sizes $\gamma_n$ and the sequence $\lambda_n$  both satisfy $\Sigma_{i=1}^{∞} \lambda_n \gamma_n < ∞ $ and $\Sigma_{n=1}^{∞} \gamma_n = ∞$, and then additionally that either (a) $\Sigma_{i=1}^{∞} \gamma_n^2 < ∞$ or (b) $\lambda_n = 0$, $K_1 = 0$, and $\lim_{n \to ∞} \gamma_n < \frac{2}{L K_2}$. Under these assumptions, they prove that $\lim_{n \to ∞} \langle \nabla f(x_{n-1}), \mathbb{E} [s_n] \rangle = 0$. In the stochastic setting, we have that $S_n = g(x_{n-1})$, and as such their assumptions reduce to: (a) An assumption similar to ES: $\mathbb{E} \left[ | g(x_{n-1}) |^2 \right] \leq \lambda_n + K_1 f(x_{n-1}) + K_2 | \nabla f(x_{n-1}) |^2$. (b) $\Sigma_{i=1}^{∞} \lambda_n \gamma_n < ∞ $ and $\Sigma_{n=1}^{∞} \gamma_n = ∞$. (c). either (1) $\Sigma_{i=1}^{∞} \gamma_n^2 < ∞$ or (2) $\lambda_n = 0$, $K_1 = 0$, and $\lim_{n \to ∞} \gamma_n < \frac{2}{L K_2}$. Part (a) would subsume our assumption if we could set $\lambda_n$ to be a constant, but the conditions they have on the sequences $\lambda_n$ and $\gamma_n$ are not the same as in our work-- we explicitly set the stepsize instead, and obtain non-asymptotic results. Two of our non-asymptotic convergence rates are novel and add to the existing literature, please see point (1) in the response to Reviewer VX3G.
>
>
> 2. (Variance-reduced stochastic gradients). While we believe that exploring variance-reduced algorithms such as SVRG or SAGA is important, our work is primarily about exploring vanilla SGD. SVRG or SAGA requires using estimators that periodically recompute full gradients, and often necessitate either mean-square average smoothness, finite-sum structure, and/or that the variance of the stochastic gradients is uniformly bounded. We do not make these assumptions in this work, and as such cannot expect to get a benefit with variance-reduction. In general, mean-squared smoothness does not imply ES (see point 3 in our comments to Rev. kLwJ), and  After our work, another paper (Zhizhe Li and Peter Richtárik, A Unified Analysis of Stochastic Gradient Methods for Nonconvex Federated Optimization, 2022 arXiv 2006.07013) carried out an analysis of variance-reduced algorithms under a generalization of our assumption plus mean-squared smoothness.
>
> 3. (Organization of the paper). We believe that one of the main contributions of the paper is to enumerate some of the settings under which ES holds, and we spend 3 pages before the convergence theory doing that-- this is because our argument is that the assumption we introduce captures different problems well. We also review prior work at length because there has been a lot of work on finding assumptions under which SGD works, and we wanted to organize it in a way the community might find useful. In this manner, the choice of putting the discussion of convergence last is intentional. We are open to adding more details in the convergence theory section if there are any that you believe may be useful to include to the reader.

---

### Decision · Action_Editors · 2023-01-04

**Recommendation:** Accept as is

**Comment:**

The paper is well-written and gives a comprehensive overview of the existing literature elaborating on connections between the existing assumptions and the proposed model here. The authors do an excellent job of demonstrating the utility of their model with numerous applications to various practical settings. There was a bit of concern shared by reviewers regarding the novelty of the techniques but the overall picture the paper contributes in furthering our understanding of SGD for large scale nonconvex problems far outweighs those concerns. The response from authors did a good job of clarifying some of the questions that came up during the review process.

**Audience:**

The intended audience for this paper includes both theoreticians and practitioners interested in principled methods for large-scale machine learning.

**Claims And Evidence:**

The paper focuses on understanding the complexity of stochastic gradient descent (SGD) for large-scale nonconvex optimization problems that arise in modern machine learning. In particular, the authors focus on studying the complexity of SGD for finding $\epsilon$-stationary points of a smooth nonconvex function. The authors introduce a new assumption for modeling the second moments of the stochastic gradients, which they call expected smoothness. Expected smoothness is more general than the existing models, including assumptions such as bounded variance, maximal/expected strong growth condition, relaxed growth, gradient confusion, and sure-smoothness condition. The authors recover the optimal rate for finding a stationary point of general smooth nonconvex problems and for finding a global solution under the PL condition. The paper is well-written and gives a comprehensive overview of the existing literature elaborating on connections between the existing assumptions and the proposed model here. The authors do an excellent job of demonstrating the utility of their model with numerous applications to various practical settings and corroborating their theoretical results with an empirical study.

Overall, a good paper. I recommend that it be accepted to TMLR. I encourage the authors to incorporate the feedback from the reviewers as they work toward the final version.